# A versatile information retrieval framework for evaluating profile strength and similarity

Alexandr A. Kalinin [1], John Arevalo [1], Erik Serrano[2], Loan Vulliard [3], Hillary Tsang[1], Michael Bornholdt[1], Alán F. Muñoz[1], Suganya Sivagurunathan [1], Bartek Rajwa[4], Anne E. Carpenter [1], Gregory P. Way [2] ✉ & Shantanu Singh [1] ✉

Large-scale profiling assays capture a cell population's state by measuring thousands of biological properties per cell or sample. However, evaluating profile strength and similarity remains challenging due to the high dimensionality and non-linear, heterogeneous nature of measurements. Here, we develop a statistical framework using mean average precision (mAP) as a single, data-driven metric to address this challenge. We validate the mAP framework against established metrics through simulations and real-world data, revealing its ability to capture subtle and meaningful biological differences in cell state. Specifically, we use mAP to assess a sample's phenotypic activity relative to controls, as well as the phenotypic consistency of groups of perturbations (or samples). We evaluate the framework across diverse datasets and on different profile types (image, protein, mRNA), perturbations (CRISPR, gene overexpression, small molecules), and resolutions (single-cell, bulk). The mAP framework, together with our open-source software package *copairs*, is useful for evaluating high-dimensional profiling data in biological research and drug discovery.

Today, the study of complex diseases and biological processes at the systems level increasingly relies on the use of multiplex and high-throughput experiments. One particular experimental design, known as "profiling," has emerged as a powerful approach to characterize biological functions, classify patient subpopulations, and identify promising therapeutic targets[1–6]. A typical profiling experiment measures hundreds to tens of thousands of features of a biological system simultaneously across many samples. The measurements can report on bulk properties or offer single-cell resolution, depending on the experimental design and research question asked. Thus, they convey information about the molecular (e.g., genomic, epigenomic, transcriptomic, proteomic, or metabolomic) or cellular (e.g., morphological, spatial, viability across cell lines) status of the system. In some profiling experiments biological samples are subjected to various perturbations, usually chemical compounds or genetic reagents[6–13].

Ultimately, by combining high-dimensional readouts, diverse biological samples, and a large number of perturbations, profiling can reveal the mechanisms of biological processes and potential therapeutic avenues.

By casting a wide and systematic net, profiling experiments provide a rich source of information for elucidating molecular and cellular responses to perturbations through comparing their readout signatures. Profiling data analyzes enable functional annotation of uncharacterized perturbations, identification of perturbation groups using clustering, and visualization of complex relationships with dimensionality reduction techniques[2–6,11]. Due to the variation in responses to different perturbations, it is crucial to prioritize perturbations that exhibit strong and reproducible phenotypic effects that are more likely to reflect true biological signals. Likewise, it is important to be able to distinguish treatments that produce cells that

[1]Imaging Platform, Broad Institute of MIT and Harvard, Cambridge, MA, USA. [2]Department of Biomedical Informatics, University of Colorado School of Medicine, Aurora, CO, USA. [3]Systems Immunology and Single-Cell Biology, German Cancer Research Center (DKFZ), Heidelberg, Germany. [4]Bindley Bioscience Center, Purdue University, West Lafayette, IN, USA. ✉e-mail: gregory.way@cuanschutz.edu; shantanu@broadinstitute.org

genuinely resemble each other (such as chemicals with the same mechanism or genes with the same function) and those that appear similar due to biological variation and confounding factors. However, the variety of perturbations, high dimensionality of readouts, and the overall heterogeneous nature of profiling datasets make it challenging to discern biologically meaningful patterns from noise and technical variation[14–19]. The ability to systematically prioritize perturbations for downstream analysis, optimize preprocessing and experimental design choices, and evaluate profile similarity is essential for maximizing the utility of these datasets.

Unlike differential feature analysis that aims to identify individual readouts that differ between samples[20], profiling data analysis treats readout signatures as holistic representations that comprehensively reflect the cellular state[2,4,5,15]. The most commonly used methods for evaluating profile strength and similarity are either based on statistical testing or, more recently, on machine learning (ML)[15,16]. Traditional multivariate statistical tests such as MANOVA and Hotelling's $T^2$ along with other recent parametric approaches[21,22] still rely on assumptions that each feature's measurements are normally distributed, sample sizes are larger than feature space dimensionality, and that, for the most part, observed phenomena are linear and not co-dependent[15,23]. Thus, these approaches broadly oversimplify the behavior of biological systems[24]. Multivariate nonparametric kernel tests[25] still assume sufficient sample size to obtain informative embeddings of distributions and require careful choice of the kernel. On the other hand, ML strategies use a classifier to sort measured phenotypes into distinct groups, where biological replicability and activity are determined by a better ability to classify samples from controls or each other. However, these methods are not readily adopted by the community for this purpose, because in addition to the high computational cost of creating numerous pairwise classifiers, ML strategies face the dual challenges of limited replicates in biological studies and overfitting, which requires extensive model evaluation, analytical design (e.g., model selection, train/test splitting), and parameter tuning. Additionally, ML approaches can overfit confounding variables, such as batch effects, which can be hard to ascertain, especially if the model is not fully explainable[26]. This precludes the selection of a single set of parameters that applies to all scenarios, because these choices are (and should be) influenced by each particular experimental design or application. Data-driven profile evaluation that does not require extensive parameter tuning and adapts easily across various experimental designs is therefore preferred.

To overcome these issues, we proposed to approach profile evaluation as an information retrieval problem and developed a statistical framework and open-source software for retrieval-based assessment of profile strength and similarity. Specifically, we employ mean average precision (mAP) as a single data-driven evaluation metric that we adapt to multiple useful tasks in profiling analysis, including determining the similarity of perturbations or groups of perturbations to controls and/or to each other. mAP assesses the probability that samples of interest will rank highly on a list of samples rank ordered by some distance or similarity metric. With appropriate distance metric choice, mAP is inherently multivariate, nonparametric, and does not make linearity or sample size assumptions, unlike most commonly used alternatives. We provide a detailed description of mAP properties in this context and a method for assigning statistical significance to mAP scores such that resulting p-values can be used to filter profiles by phenotypic activity and/or consistency. We show the advantages of mAP over existing metrics using simulated data and illustrate the utility of mAP on a variety of real-world datasets, including image-based (Cell Painting[27]), protein (nELISA[12]), and mRNA (Perturb-seq[7–10]) profiling data, some at the single-cell level, and involving several perturbation types (CRISPR gene editing, gene overexpression, and small molecules). We provide a Python package *copairs* implementing a flexible framework for grouping profiles based on metadata, and

efficient calculation of mAP scores and corresponding p-values for easy and scalable application of our method to other datasets. We expect that the mAP framework we provide will streamline hypothesis generation and improve hit prioritization from a wide range of large-scale, high-throughput biological profiling data.

## Results

### Profile evaluation as information retrieval

A fundamental goal of profiling analysis is to identify biologically meaningful relationships between samples by comparing their phenotypic signatures. One important application of this is the ability to annotate previously uncharacterized perturbations by comparing them to a reference dataset of annotated profiles (or "compendium")[1,2,4,6]. For example, a compound with an unknown mechanism of action (MoA) can be compared against a compendium of compounds with known MoAs. If the unknown compound exhibits a phenotypic signature highly similar to those of compounds targeting a specific pathway, it suggests a shared mechanism and potential therapeutic relevance.

This problem naturally aligns with principles of information retrieval, where the goal is to rank and retrieve relevant items from a large dataset (annotated perturbation profiles) based on their similarity to a given query (uncharacterized perturbation profile)[28]. Similarity-based retrieval allows evaluating how similar a query profile is to a given group (e.g., a shared MoA) in a data-driven manner, without making assumptions about the distribution or interpretability of features, linear separability, or the number of reference samples per group.

In high-throughput profiling experiments, perturbations vary widely in their effects, from strong, reproducible phenotypic changes to those indistinguishable from controls. To ensure that retrieval-based annotation is meaningful, both the reference compendium and the uncharacterized profiles should exhibit robust phenotypic signals. Similarity-based retrieval also provides an effective way to evaluate this task by assessing whether a perturbation reliably retrieves its own replicates over controls—a task we refer to as *phenotypic activity*—only requiring the minimum of two replicates per perturbation. By filtering out phenotypically inactive perturbations, we can focus on perturbations that induce biologically relevant effects. Additionally, if most perturbations (or known strong perturbations) fail to retrieve their own replicates or are indistinguishable from controls, this may indicate systemic issues with the whole dataset that can prevent detecting real effects. For phenotypically active perturbations with known annotations, retrieval-based assessment can also be used to evaluate *phenotypic consistency*—the degree to which perturbations with a shared annotation (e.g., a known MoA) exhibit a distinct and cohesive signature compared to other groups. Assessment of phenotypic consistency in profiling data using mAP can help prioritizing perturbation groups that produce robust phenotypes but also find useful and previously unrecognized connections. Finally, we also can assess *phenotypic distinctiveness* of an active perturbation by retrieving its replicates against all other active perturbations. Because all these tasks are framed as information retrieval (Supplementary Table 1), we can leverage a single retrieval-based metric (mean average precision, mAP) for their evaluation. Hence, we refer to our approach to profile evaluation as *the mAP framework*.

By assessing both phenotypic activity and consistency through information retrieval, we can enhance the ability to distinguish meaningful biological relationships from artifacts of experimental noise or technical variation in downstream analyses.

### The mAP framework overview

In this section, we demonstrate through a simple example the application of the proposed mAP framework to evaluating phenotypic activity of a perturbation (Fig. 1).

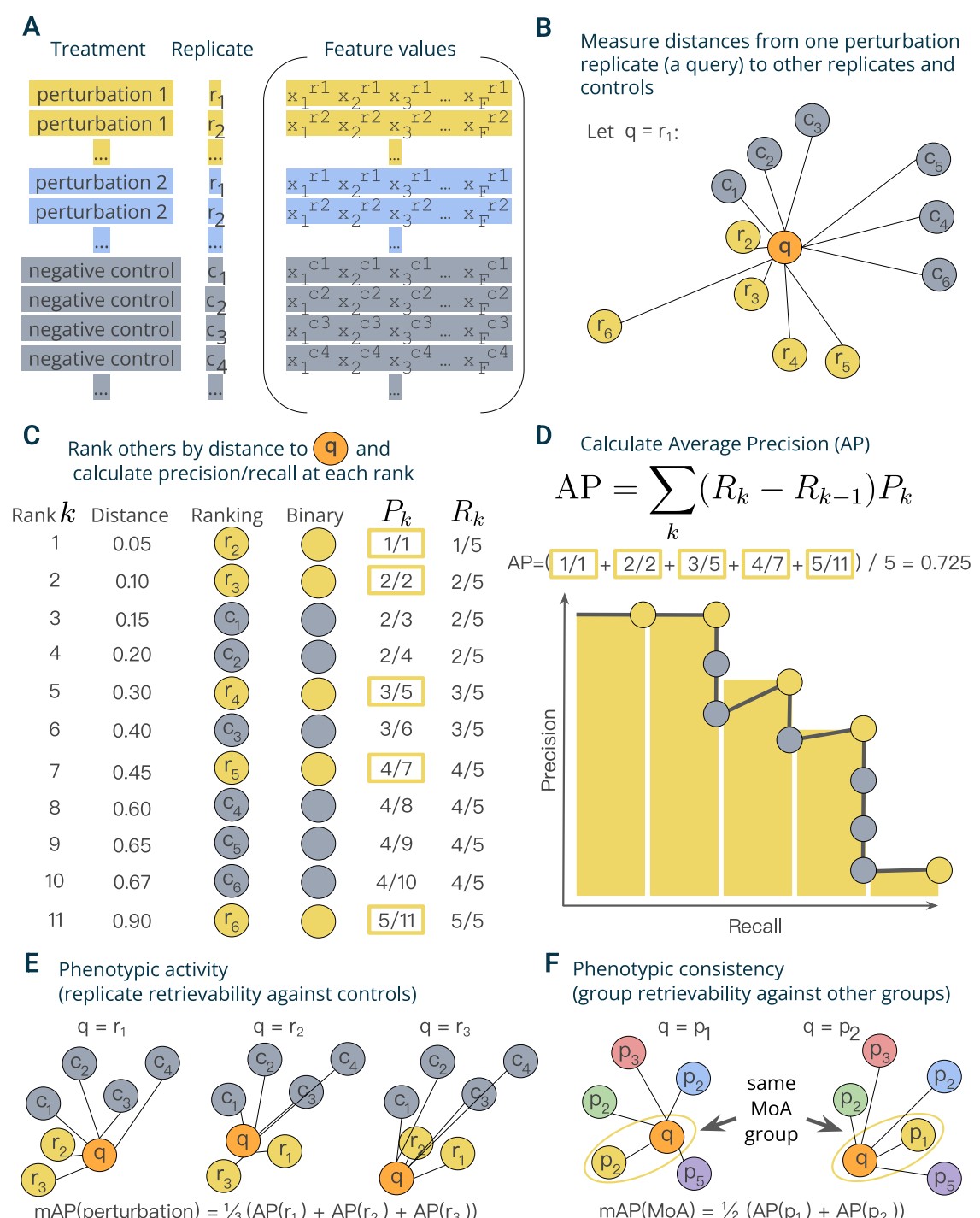

**Fig. 1 | Schematic overview of the mAP framework. A** A typical output of a profiling experiment contains multiple replicate profiles for each perturbation and controls. **B** To measure average precision (AP) per perturbation replicate, we selected one replicate profile as a query and measured distances to its other replicates and controls. **C** Profiles were then ranked by decreasing similarity (increasing distance) to the query; the rank list was converted to binary form and used to calculate precision $P_k$ and recall $R_k$ at each rank $k$. **D** Average precision was calculated by averaging precision values over those ranks $k$ containing perturbation replicates, which corresponds to a non-interpolated approximation of the area under the precision-recall curve. **E** By applying this procedure to each perturbation replicate, we calculated a set of AP scores that were then averaged to obtain a mAP score for a perturbation's *phenotypic activity*. **F** One can also apply the same framework to retrieving groups of perturbations with the same biological annotations (rather than groups of replicates of the same perturbation)—for example, compounds that share the same mechanism of action (MoA)—by calculating the mAP score per each group of perturbations (MoA). Source data are provided as a Source Data file.

Mean average precision (mAP) is a performance metric routinely used in information retrieval and machine learning, particularly in the context of ranking tasks[28,29]. mAP measures the ability to retrieve samples within a group ("correct" samples) from a collection of samples from another group ("incorrect" samples). We used mAP to indicate the degree to which profiles from one group exhibit greater intra-group similarity compared to their similarity with the profiles from a second group.

In a typical profiling experiment, both perturbations and controls are represented by multiple biological replicates, i.e., the same

perturbation is replicated across multiple wells and, depending on the experiment's scale, even multiple plates and batches (Fig. 1A). These replicate profiles can be obtained directly at the well level or first measured at the single-cell level and then aggregated. We illustrate calculation of mAP for *phenotypic activity* assessment through retrieving a group of perturbation replicate profiles against a group of control replicate profiles. This example demonstrates the simplest block design, when we group profiles only by the replicate identity and not other metadata variables, such as well position, plate, etc (Fig. 1A). In experimental design, a block design refers to the arrangement of experimental units into groups (blocks) that are similar to one another, to reduce sources of variability and increase the precision of the comparisons being made[30]. The choice of block design can impact the mAP calculation, as it determines which profiles are considered "correct" or "incorrect" when evaluating retrieval performance. As discussed later, more complex block designs can be used to account for the impact of other sources of variation (e.g., well position or plate effects) on the retrieval task of interest by grouping profiles accordingly.

Profiles can be viewed as points in a high-dimensional feature or representation space, where the closeness between pairs corresponds to profile similarity. Profile similarity can be assessed by measuring the distance between these points defined by any relevant distance function, such as cosine, Euclidean, Mahalanobis, etc., such that a larger distance between points indicates lower similarity and vice versa. Following a typical information retrieval workflow[28], we began by designating one profile from a replicate group as a query and measuring distances between the query and the rest of this perturbation's replicates as well as control replicates (Fig. 1B).

We rank profiles by their decreasing similarity to the query, such that the most similar profile is at the top of the list (Fig. 1C). We then convert this ranked list to a binary form by replacing perturbation replicates with ones (these are "correct matches", i.e., expected to be more similar to the query) and controls with zeros (they are" incorrect matches", i.e., expected to be less similar to the query). In an ideal scenario where a perturbation produces a strong signal that is technically replicable, all perturbation replicates are more similar to each other than to controls and, hence, will appear on the top of the list. However, in practice, it is often challenging to detect differences from controls, especially given the presence of technical variation (Supplementary Fig. S1). Having a binary rank list allows calculating precision and recall at each rank $k$. Precision at rank $k$ (also called Precision@k) is the fraction of ranks 1 to $k$ that contain correct matches (ones). Recall at rank $k$ is the fraction of the correct matches (ones) across ranks from 1 to $k$.

Although there are multiple possible ways to aggregate precision and/or recall values, we chose to calculate average precision (AP) because of its statistical properties. It has an underlying theoretical basis as it corresponds to the area under the precision-recall curve[29], it allows a probabilistic interpretation[31], it has a natural top-heavy bias[32] (top-ranked correct matches contribute more than low-ranked), it is highly informative for predicting other metrics such as R-precision[33], and finally, it results in good performance when used as an objective in learning-to-rank methods[34]. Although many formulations of AP exist[35], we calculate the conventional non-interpolated AP score as the average value of precision over those ranks $k$ that contain correct matches[29] (Fig. 1D).

By sequentially using each replicate as a query to retrieve the remaining replicates, we calculate replicate-level AP scores (Fig. 1E). These scores can identify outlier replicates that deviate substantially from their group. Averaging these scores yields the perturbation-level mean Average Precision (mAP) score[29]. This score effectively quantifies the *phenotypic activity* of a perturbation, reflecting the average extent to which its replicate profiles were more similar to each other compared to control profiles (Fig. 1E). We also calculate mAP to assess the

*phenotypic consistency* of multiple group members annotated with common biological mechanisms or modes of action (Fig. 1F). In this setting, we first aggregate replicate profiles at the perturbation level (for example, by taking the median value for each feature of single cells). We then apply the mAP framework to quantify to what extent perturbations with related biological annotations produce profiles that resemble each other compared to other perturbations in the experiment. By using other perturbations for the null distribution instead of negative controls, we assess the biological specificity of each group of profiles relative to other samples.

The mAP framework is implemented such that it can accommodate any distance measure that accepts a pair of points and returns their (dis)similarity, allowing for customization best suited for the particular type of data at hand. The careful choice of the distance metric is crucial and may vary across profiling experiments. In this article, we used cosine distance due to its ability to identify related samples from biological perturbational data based on the similarity of their change patterns, rather than the extent of these changes[36–38]. As a similarity-based approach, mAP is not immune to challenges typical for the analysis of profiling data that can be noisy and sparse. To enhance the observation of useful biological patterns, profile preprocessing may include[15,39]: sample and feature filtering, missing value imputation, dimensionality reduction, feature transformation, normalization, and selection.

## Statistical significance of mAP

In all cases, we determined the statistical significance of the mAP score using permutation testing, a common method for significance determination when distribution of the test statistic is unknown (for example, not known to follow a normal distribution). This approach is frequently applied in biological data analysis, including high-throughput screening[22]. Under the null hypothesis, we assume that both perturbation and control replicates were drawn from the same distribution. We generate mAP distribution under the null hypothesis, by repeatedly reshuffling the rank list and calculating mAP. The *p*-value is then estimated according to standard practices for permutation-based methods, defined as the fraction of permutation-derived mAP values that are greater than or equal to the original mAP value. This approach aligns with the interpretation of significance values in parametric statistical analyzes, where a nominal significance cutoff of 0.05 is typically used. Finally, these *p*-values are corrected for multiple comparisons using False Discovery Rate (FDR) control methods, such as Benjamini–Hochberg procedure[40] or its alternatives[41]. We refer to the percentage of samples with calculated mAP scores having a corrected *p*-value below 0.05 as the *percent retrieved* (see "*Methods: Assigning significance to mAP scores*" for details). Very low values of percent retrieved can indicate widespread assay insensitivity, suboptimal feature extraction, or uncontrolled experimental variation.

We therefore concluded that the mAP framework, as described and applied, could assess various qualities of high-dimensional profiling data by quantifying the similarity within a group of profiles in contrast to their similarity to another group. Unlike existing solutions, mAP is completely data-driven, does not involve complex calculations or parameter tuning, and is independent of the underlying nature of the observations given the appropriate selection of the distance measure. It is flexible across various experimental designs and offers a robust means to ascertain the statistical significance of the observed similarities or differences.

## mAP detects profile differences introduced in simulated data

We next sought to rigorously assess and compare our mAP framework in phenotypic activity assessment against existing metrics using simulated data, where profile characteristics could be carefully controlled. Among established approaches, we selected the multidimensional perturbation value[22] (mp-value) for comparison because

it is multivariate by design, can be applied to any two groups of profiles, and has been shown to outperform other approaches, including univariate and clustering-based, in a simulation study with a similar design[22]. It is based on a combination of principal component analysis, Mahalanobis distance, and permutation testing to determine whether two groups of profiles differ from each other. Another method we choose to compare with mAP is maximum mean discrepancy test (MMD)[25], which is a nonparametric kernel-based method for multivariate two sample testing. Finally, we also directly clustered profiles using the k-means algorithm, which was considered to be successful when it correctly separates perturbation and control profiles. By comparing our framework with these established methods in controlled scenarios that mimic real-world experimental designs, we aimed to evaluate mAP's potential as a more effective tool for analyzing differences in profiling data.

We conducted simulations to evaluate mAP performance by generating perturbation and control profiles such that each perturbation had 2 to 4 replicates, in experiments with 12, 24, or 36 control replicates (in all combinations). The simulated profiles varied in feature size, ranging from 100 to 5000, representing a typical range for various kinds of profiles, particularly after feature reduction or selection. We generated control profile features using a standard normal distribution $\mathcal{N}(0,1)$. For perturbation replicates, a certain proportion of features (ranging from 1% to 64%) were sampled from a normal distribution with a shifted mean $\mathcal{N}(1,1)$, while the rest were drawn from $\mathcal{N}(0,1)$. Following the previously described method, we calculated mAP scores and corresponding $p$-values at the perturbation level to assess phenotypic activity for each perturbation (see Fig. 1E). We measured performance by calculating recall as the proportion of the 100 simulated perturbations for which each metric achieved statistical significance ($p < 0.05$).

mAP consistently detected the same or higher proportion of perturbations compared to mp-value, MMD, and k-means clustering in most simulated scenarios (Fig. 2, Supplementary Fig. S2). Our findings highlighted that all metrics were sensitive to the experimental design, including the number of replicates and controls, the dimensionality (number of features) of the dataset, and the proportion of features that were perturbed. As expected, a decrease in the number of replicates and controls generally led to reduced performance for all metrics. mAP's recall rate consistently improved with an increase in the number of features and the proportion of perturbed features. This trend highlights mAP's adaptability to high-dimensional data, a critical advantage in handling the vast feature spaces typical in modern profiling assays. In contrast to mAP and k-means clustering, mp-value and MMD were less stable and often demonstrated stagnation or decline in recall with an increase in the number of features. These results were further confirmed by additional simulations that used a normal distribution with other parameters (Supplementary Fig. S3) and with a more challenging heavy-tailed Cauchy distribution (Supplementary Fig. S4). Finally, testing mAP with Pearson correlation and Euclidean distance as a similarity metric showed performance competitive with alternative metrics, albeit lower compared to our default choice of cosine distance (Supplementary Fig. S5). All methods struggled to reach a retrieval rate of 20% when perturbed profiles differed from controls in only a few features. To alleviate this issue, we recommend dimensionality reduction and feature selection to improve signal-to-noise ratio prior to profile evaluation[15]. While in theory it is possible to test for differences in individual features using mAP, it would not be practical, because the main purpose of this framework lies in high-dimensional profile similarity analysis; other methods are more suited to analyzing individual features[20].

Taken together, our findings reveal mAP's consistent performance in most scenarios, highlighting its potential as an effective and adaptable tool for biological data analysis compared to existing methods. Specifically, we found mAP could sensitively detect subtle differences between samples, in the context most relevant to large high-dimensional profiling datasets: scenarios when the number of features was much larger than the number of replicate profiles per sample. While approaches such mp-value and MMD aim to represent and compare distributions estimated from a very few high-dimensional observations, mAP achieves better sensitivity by relying on discretized ranking of pairwise distances. It is simpler and more efficient, without requiring complex matrix operations needed for calculating mp-value and MMD, and without multiple restarts needed to reach a robust solution for k-means.

## mAP captures diverse properties of real-world morphological profiling data with both genetic and chemical perturbations

Next, we demonstrated the versatility of the mAP framework through its application to different tasks on real-world data, evaluating the effects of selected preprocessing methods and experimental designs. We began with image-based profiles of genetic perturbations and tested several ways mAP can be used for tasks beyond ranking perturbations by their phenotypic activity. We chose our published Cell Health dataset of Cell Painting images of CRISPR-Cas9 knockout perturbations of 59 genes, targeted by 119 guides in three different cell lines[42]. We used a subset of 100 guides that had exactly six replicates (two replicates in three different plates) in each cell line.

We used mAP to evaluate **phenotypic activity** (replicate retrievability against non-targeting cutting controls[43]) (Fig. 1E) for four different tasks (Fig. 3A–D). First, we assessed the overall quality of the dataset by checking that at least some guides resulted in phenotypes robustly distinguishable from controls. Second, we compared how two different data preprocessing methods influence the effects of technical variability on the phenotypic activity of each guide across cell lines. Third, we ranked individual guides by phenotypic activity and filtered out inactive ones for downstream analysis of phenotypic consistency. Finally, we compared the contributions of each fluorescent channel to guide phenotypic activity.

First, we calculated mAP for guide phenotypic activity using two different data preprocessing methods (Fig. 3A). The first preprocessing method included data standardization by subtracting means from feature values and dividing them by standard deviation using the whole dataset. Alternatively, we used a robust version of standardization, which replaces mean and variance with median and median absolute deviation, correspondingly, and is applied on a per-plate basis ("MAD robustize"). In each scenario, we retrieved grouped profiles against negative controls from both plates and reported percent retrieved (percentage of mAP scores with a corrected $p$-value below 0.05). For both preprocessing methods and all three cell types, retrieval percentages ranging 13%-94% indicated presence of perturbations with distinguishable phenotypes, and the full range of mAP values showed that perturbation effects varied from weak to very strong, confirming the good overall dataset quality.

Then, we leveraged the fact that each guide had replicates in different well positions and plates to formulate three profile groupings for well position and plate effect assessment (Fig. 3A). The first group only included profiles derived from different plates and well position; the second group only included profiles from the same well position, but different plates; and the third group only included profiles from the same plate, but different wells. Our framework implementation allows specifying which metadata columns should have the same or different values for a pair of profiles to belong to the same group. In the absence of well-position and plate effects on phenotypic activity, all three tasks should demonstrate similar mAP values. However, as expected, the data reveals differences in retrieval rates due to technical variation (Fig. 3A): with standardized profiles, retrieval of replicates in a different well position and different plate had the lowest scores (28% retrieved on average across cell lines), while sharing the same well position or plate resulted in higher scores (46% and 44%

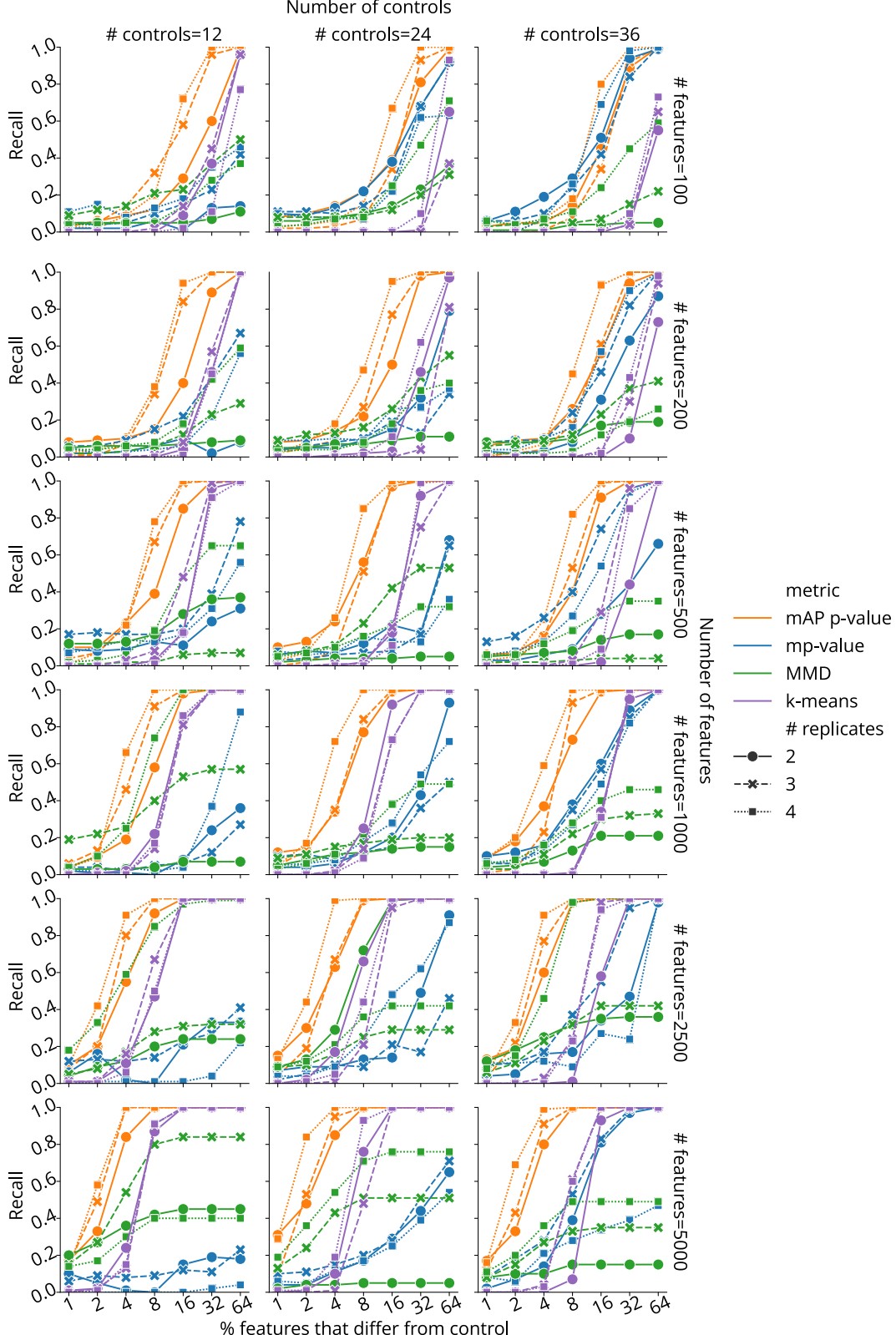

**Fig. 2 | The mAP framework evaluation on simulated data.** Benchmarking retrieval performance of mAP $p$-value (orange), mp-value (blue), MMD $p$-value (green), and k-means clustering (purple) for retrieving phenotypic activity on simulated data, where unperturbed and perturbed features are sampled from $\mathcal{N}(0,1)$ and $\mathcal{N}(1,1)$, correspondingly. Recall indicates the percentage of 100 simulated perturbations under each condition that were called accurately by each method (as distinguishable from negative controls, or not). The horizontal axis probes what proportion of the features in the profile were different from controls (note the binary exponential scaling). Marker and line styles indicate different numbers of replicates per perturbation (# replicates of 2, 3, and 4). Columns correspond to the different number of controls (# controls of 12, 24, and 36). Rows correspond to different profile sizes (# features being 100, 200, 500, 1000, 2500, and 5000). mAP, mp-value, and MMD used a one-sided permutation test to obtain $p$-values without adjusting for multiple comparisons; no statistical test was performed for k-means. Source data are provided as a Source Data file.

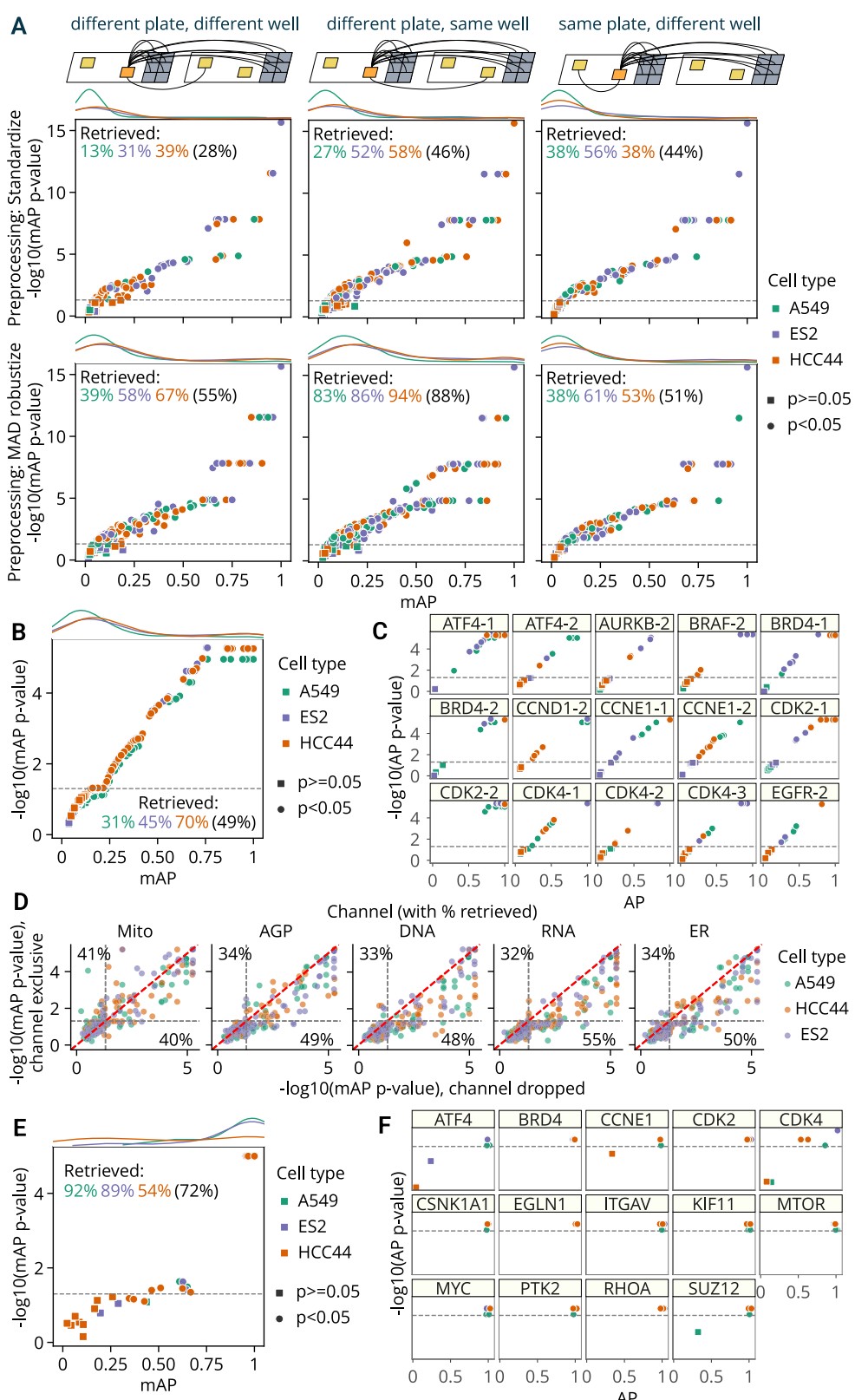

retrieved on average, correspondingly). Using standardization per plate, results for A549 and ES2 cell lines had higher retrieval rates when considering replicates from the same plate (38% and 56% correspondingly) vs not sharing the same plate (13% and 31%), indicating the presence of plate-to-plate variability. For the HCC44 cell line, retrieval rate on the same plate (38%) was not better than for replicates across plates (39%). By contrast, using robust standardization (MAD robustize) per-plate increased retrieval of profiles from a different well position and different plate (55% retrieved) to a larger extent than it did for the same plate, different well test (51% retrieved). But it also inflated retrieval of profiles that share the same well position in different plates (88% retrieved), demonstrating that well position effects were not addressed by this pre-processing and may affect downstream analyzes. These results were observed on the level of individual cell lines as well,

**Fig. 3 | The mAP framework applied to morphological profiling of CRISPR-Cas9 knockout perturbations (Cell Health dataset). A** mAP is calculated to assess well position and individual plate effects on phenotypic activity by retrieving guide replicates against controls in three scenarios (replicates of a guide across different plates and well positions; replicates of a guide across different plates, but in the same well position; and replicates of a guide within the same plate, but in different well position) and two data preprocessing methods (standardize and MAD robustize per plate). Percentages retrieved indicate the percentage of scores with *p*-value below 0.05 per cell line (and averaged across all cell lines in parenthesis). **B** mAP is calculated to assess the phenotypic activity of perturbations by guide replicate retrievability against controls in three cell lines individually (49% retrieved on average across all three cell lines). Results included all three replicate plates available per cell line. **C** Replicate-level AP scores calculated for a subset of guides from

(**B**) highlight the variation from guide to guide across cell lines. **D** mAP *p*-values estimated to assess the influence of individual fluorescence channels on guide phenotypic activity against controls by either dropping a channel or including only that single channel (percent retrieved is shown for each axis); these results can be compared to 49% retrieved when all channels' data is available (on average across all three cell lines, as in **B**). **E** mAP is calculated to assess the phenotypic consistency of guides annotated with related target genes (against guides annotated with other genes) in three cell lines individually. **F** Guide-level AP scores calculated for a subset of genes from (**E**) highlight the variation from gene to gene across cell lines. mAP *p*-values were estimated using a one-sided permutation test and adjusted for multiple comparisons by Benjamini–Hochberg procedure. Percent retrieved indicates the percentage of scores with *p*-value below 0.05. Source data are provided as a Source Data file.

with all three having similar retrieval rates independently of sharing a specific plate but showing substantially higher rates when sharing a well position (Fig. 3A). We used MAD robustize to preprocess for all subsequent analysis given its better performance on a challenging task (retrieving from different well position, different plate). This example showcased the flexibility of the mAP framework for grouping profiles according to experimental properties and assessing impact of technical variation in profiling data on phenotypic activity of perturbations. However, we note that evaluating batch effects and methods for their correction is a complex problem that may require using multiple specialized metrics for a comprehensive assessment[17–19].

Next, we used all six replicates per CRISPR guide to assess its phenotypic activity (Fig. 1E) by retrieving each perturbation's replicates against non-targeting controls in three cell types. Retrieval percentage varied 31-70% by cell line, respectively (Fig. 3B), showing that mAP captures cell context-dependent differences of each guide's phenotypic activity, though potentially confounded by well-position and plate effects. We also showed cell line-dependent differences in individual replicate AP scores for a subset of guides (Fig. 3C). For example, while five out of six replicates of the *ATF4-1* guide in ES2 cells showed high similarity and clear distinction from controls, the sixth replicate did not, as indicated by its low AP score and high p-value, suggesting it may be an outlier. We observed similar retrieval rates using an alternate negative control, wells that were not perturbed at all (Supplementary Fig. S6A). The significance of mAP was somewhat negatively correlated with CERES scores[44] (Supplementary Fig. S6B), a measure of gene essentiality derived from viability experiments, confirming that many perturbations that impact viability also impact morphology[45], though one would expect many exceptions, for example, for genes that are not expressed well in the given cell type. This assessment allowed us to filter out "inactive" guides that produce phenotypes indistinguishable from controls to ensure that at the next step, perturbation similarities are not due to shared lack of activity.

Then, we applied the mAP framework to characterizing contributions of different fluorescence channels by calculating metrics for each single channel individually (Fig. 3D, Y axis); the mitochondria channel proved the most independently useful for retrieving guide replicates against controls. In most cases, dropping a channel (Fig. 3D, X axis) only slightly diminished retrieval performance, a useful guide for researchers wanting to swap out a channel for a particular marker of interest. In a similar fashion, we assessed the contributions of different feature types extracted from different cell compartments and found (Supplementary Fig. S6C) that, for example, excluding Radial-Distribution or AreaShape features dropped the percentage of retrieved guides to below 35%. Removing Texture or Intensity features resulted in ~70% retrieval rates, which can hint at what phenotypic responses were distinguishing for this gene set as a whole.

We next assessed **phenotypic consistency** of CRISPR guides that targeted the same gene by retrieving them against guides that targeted other genes (similar to Fig. 1F), to see whether guides targeting the same gene yielded a consistent and relatively distinctive phenotype.

First, we aggregated each guide's six replicates by taking the median value for each feature. Then, we filtered guides that did not pass the significance threshold for phenotypic activity in each cell type (Fig. 3B) to remove profiles that could not be confidently distinguished from controls. There were two aggregated guide profiles per gene annotation, which we retrieved against guide profiles of other genes (2 "replicates" vs 118 "controls" using the terms of Fig. 2). Retrieval percentages ranged from 54–92% across cell lines (Fig. 3E). We also reported per-guide AP scores for a subset of individual genes (Fig. 3F), demonstrating gene-to-gene differences and variability in guide consistency across the three cell lines. For instance, in the case of *CDK4*, A549 and HCC44 cell lines each had one guide that was inconsistent with other guides targeting the same gene.

Finally, we applied the mAP framework to **other perturbation types** (small molecules and gene overexpression, rather than CRISPR-Cas9 knockouts), to assess their phenotypic activity and consistency (Supplementary Fig. S7). We used the dataset "cpg0004"[11], which contains Cell Painting images of 1,327 small-molecule perturbations of A549 human cells and the JUMP Consortium's "cpg0016[orf]" dataset[46] of U2OS cells treated with 15,136 overexpression reagents (open reading frame - ORFs), encompassing 12,602 unique genes, including controls, making it the largest dataset in this study in terms of number of perturbations. In both cases, we first calculated mAP to assess the phenotypic activity of each perturbation by replicate retrievability against controls, which resulted in 34% of small molecules retrieved for cpg0004 (Supplementary Fig. S7A) and 56% of ORFs retrieved (Supplementary Fig. S7B). Subsequently, we filtered out perturbations that did not pass the phenotypic activity threshold and aggregated the rest on a per-perturbation basis by computing the median value for each feature across replicates. Finally, we calculated mAP to assess phenotypic consistency (a measure of whether profiles capture true biological meaning, captured here by public annotations). We tested for phenotypic consistency among small molecules that were annotated as targeting the same gene (cpg0004) or among ORFs encoding genes that produce proteins that were annotated as interacting with each other, per the mammalian CORUM database[47] (cpg0016[orf]). For cpg0004, 32% of target genes showed consistent phenotypic similarity among small molecules targeting them (Supplementary Fig. S7C); for cpg0016[orf] it was 4% of assessed protein complexes (Supplementary Fig. S7D). Evaluating phenotypic consistency by nature relies on the accuracy and completeness of external annotations. Leveraging multiple sources of annotation, such as combining pathway databases, can strengthen the interpretability of phenotypic profiling, helping to recapitulate known relationships and improving benchmarking outcomes[48]. For practical applications, incorporating diverse annotations could similarly enhance profile retrieval by allowing cross-validation of biological relationships under different contexts, such as across cell types or experimental conditions.

These results demonstrated that the proposed mAP framework can be used for assessing various properties of real-world

morphological profiling data created with both genetic and chemical perturbations. By changing how profile groupings are defined, mAP can be used for multiple purposes: to characterize technical variation in data, to evaluate methods to address them, to determine the contributions of specific fluorescent channels or measured feature types, and to ultimately select and rank perturbations by their phenotypic activity and consistency for potential downstream analyzes.

## mAP quantifies strength and similarity of protein and single-cell mRNA profiling data

To demonstrate the applicability of the mAP framework beyond image-based profiling, we applied it to other modalities, including transcriptomics and proteomics.

The first dataset contained proteomic profiles from a 191-plex nELISA, a high-throughput, high-plex assay designed for quantitative profiling of the secretome[12], which was performed in A549 cells across 306 well-characterized compound perturbations from the Broad Institute's drug repurposing library[49]. This dataset also had matching Cell Painting morphological profiles imaged from the same physical samples whose supernatants were nELISA-profiled.

First, we used mAP to assess phenotypic activity via replicate retrievability for both assays. This analysis resulted in 72% of compounds being retrieved using Cell Painting and 39% with nELISA (Fig. 4A, B); the smaller percentage is likely due to the limitations in the original experimental design that was not ideal for secretome profiling. We further calculated mAP to assess phenotypic consistency by identifying compounds annotated with the same target gene. This analysis yielded 23% retrieval for Cell Painting and 5% for nELISA (Fig. 4C, D). Similarly to phenotypic activity results, much lower percent retrieved for nELISA was likely due to A549 cells' limited secretory capacities, the absence of immune stimulation, and a mismatch between pathways targeted by small molecules and nELISA readouts[12]. This comparison validated mAP's utility in comparing two different profiling assays, offering valuable insights for planning future studies, for example, selecting an appropriate cell type for a particular assay.

Finally, we used mAP to evaluate a Perturb-seq[7–10] mRNA profiling dataset of single cells treated with CRISPRi. The experiment assessed how single-guide RNAs (sgRNAs) containing mismatches to their target sites attenuate expression levels of target genes[50]. Specifically, 25 genes involved in a diverse range of essential cell biological processes were targeted with 5–6 mismatched sgRNAs, covering the range from full to low activity, and 10 nontargeting controls. Each mismatched guide was characterized by its activity levels relative to the perfectly matched sgRNA targeting the same gene[50]. We aggregated single-cell profiles on the biological replicate level and compared mAP scores to sgRNA relative activity, expecting that guide mismatches that disrupt activity levels to a larger extent should have mRNA profiles that are less easily distinguishable from controls. We indeed observed an overall correlation between mAP scores for a sgRNA's mRNA profile similarity and its relative activity levels, (Fig. 4E), with more nuanced differences in correlations for specific genes (Fig. 4F).

These applications demonstrate mAP's robustness in quantifying the strength and similarity of image, protein, and mRNA profiles, affirming its broad utility across diverse profiling assays.

## mAP captures subtle phenotypic impacts of perturbations at the single-cell resolution

Single-cell profiling has become increasingly feasible, providing detailed and nuanced insights into the complex nature of biological systems, which are often obscured in bulk analyzes. To illustrate the mAP framework's utility in analyzing single-cell data, we applied it to two distinct single-cell profiling datasets. Because mAP is an average of AP scores calculated by using each observation as a query, it is straightforward to use single-cell AP scores to characterize individual observations and the whole query group.

First, we repeated the analysis of the Perturb-seq mRNA profiling dataset[50] (Fig. 4E, F) on the single-cell level. The overall relationships between single-cell AP scores and relative activity levels recapitulate those observed in the bulk profiles with more fine-grained details (Fig. 5A), while per-gene (Fig. 5B) and per-guide (Supplementary Fig. S8) visualizations revealed varied levels of heterogeneity across individual cells, even for guides with perfect relative activity levels.

The second dataset called "Mitocheck" contained cell images of genome-wide gene silencing by RNA interference[51]. We used a subset of these images, in which almost 3000 cells were manually annotated with observed morphological classes and processed by either CellProfiler[52] or DeepProfiler[53] feature extractors to create single-cell morphological profiles[54]. After filtering out cells that failed quality control or were out of focus, the subset contained 2456 single-cell profiles annotated with 15 morphological classes across 60 genes.

We used replicate retrievability against non-targeting controls to compare phenotypic activity of single-cell CellProfiler- and DeepProfiler-derived profiles grouped by morphological classes and target genes. Both feature extraction methods showed on average similar performance, retrieving morphological annotations with 0.33-0.42 mAP and 93-95% retrieved (Fig. 5C, Supplementary Fig. S8A). Interestingly, however, their performance varied across individual classes (Fig. 5C, Supplementary Fig. S9C), indicating some complementarity in phenotypes that are characterized more informatively with one approach than the other. Although performance was lower for both methods in the target gene retrieval task (Supplementary Fig. S9B), the decrease was more severe for CellProfiler features (0.18 mAP, 89% retrieved) compared to DeepProfiler (0.22 mAP, 92% retrieved). When comparing performance across both tasks, CellProfiler features overall demonstrated more variability across the range of mAP scores (Supplementary Fig. S9D) compared to more consistent results of DeepProfiler (Fig. 5D). Visualizing embeddings of individual morphological classes for both feature types indicated that those with higher retrieval rates resulted in more consistent clusters (Supplementary Fig. S10). Calculation of mAP $p$-values for gene phenotypic activity took the longest for this dataset due to the large size of control cells and the highly variable number of single cells per targeted gene (Supplementary Table 2).

This analysis underscores the ability of the mAP framework to discern phenotypic variability and heterogeneity inherent in single-cell data, revealing both the strengths and complementary nature of different feature extraction methodologies.

## Discussion

High-throughput profiling experiments have shown great promise in elucidating biological functions, patient subpopulations, and therapeutic targets. However, the high dimensionality and heterogeneity of profiling datasets present a significant obstacle for traditional methods in evaluating data quality and identifying meaningful relationships among profiles. Our work advances this domain by reframing profile quality assessment as an information retrieval task and by proposing a comprehensive statistical and computational framework using mean average precision (mAP) to assess profile strength and similarity. The mAP framework can be applied to image-based, protein, and gene expression profiling datasets created with either genetic and chemical perturbations. By assessing replicability of repeated experiments via retrieval of perturbation replicates against negative controls, the mAP framework checks dataset for potential dataset-scale issues, identifies phenotypically active perturbations and allows to filter out ones indistinguishable from controls. It can also be used to measure phenotypic consistency among different perturbations that share expected biological similarity, such as chemical mechanisms of action and gene-gene relationships, by retrieving perturbation groups. By selecting phenotypically active

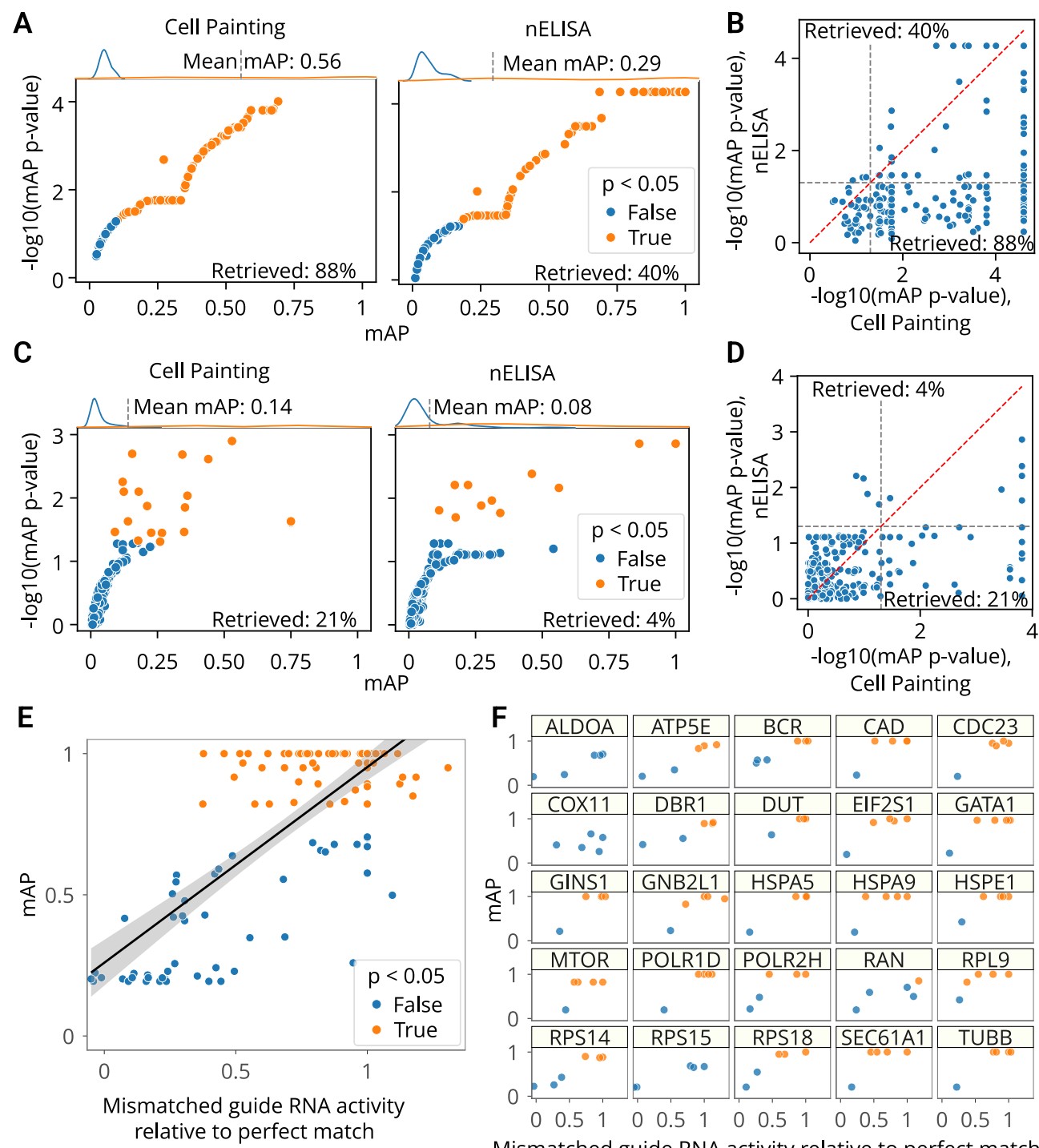

**Fig. 4 | The mAP framework applied to proteomic and mRNA profiling. A** mAP is calculated to assess the phenotypic activity of compounds by replicate retrievability against controls in matching Cell Painting and nELISA profiling data. **B** A combined view of the data from (**A**) is presented, showing phenotypic activity retrieval for both assays. **C** mAP is calculated to assess the phenotypic consistency by retrieving phenotypically active compounds annotated with the same gene target in matching Cell Painting and nELISA profiling data (note: the nELISA panel includes 191 targets including cytokines, chemokines, and growth factors which are not expected to respond well in these convenience samples from a prior study, because there is no immune stimulation and the A549 cells used have limited secretory capacity). **D** A combined view of the data from (**C**) is presented, showing

phenotypic consistency retrieval for both assays. **E** mAP is calculated to assess the mRNA profile-based phenotypic activity of a mismatched CRISPRi guide from a Perturb-seq experiment (y-axis) and correlate it with the guide's activity relative to a perfectly matching guide for that gene (x-axis). A linear model fit is shown in black with gray error bands showing the 95% confidence interval. **F** A subset of the data from (**E**) is presented, with several genes highlighted individually to demonstrate the variation from gene to gene. mAP p-values were estimated using a one-sided permutation test and adjusted for multiple comparisons by Benjamini–Hochberg procedure. Percent retrieved indicates the percentage of scores with p-value below 0.05. Source data are provided as a Source Data file.

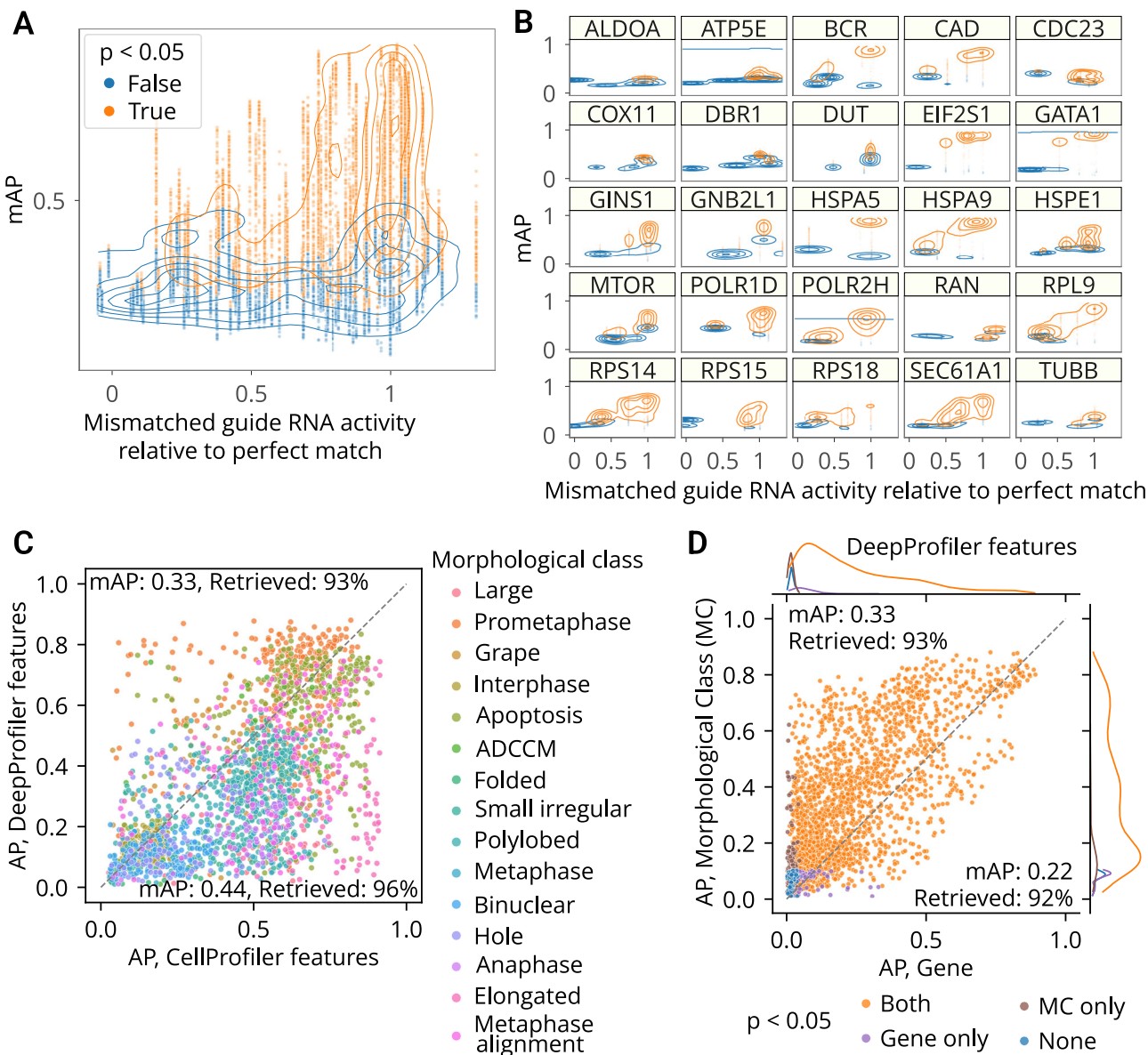

**Fig. 5 | The mAP framework applied to single-cell mRNA and imaging data. A** AP scores are calculated to assess the single-cell mRNA profile-based phenotypic activity of a mismatched CRISPRi guide from a Perturb-seq experiment (y-axis) and correlate it with the guide's activity relative to a perfectly matching guide for that gene (x-axis). **B** A subset of the data from (**A**) is presented, with several genes highlighted individually to demonstrate the variation from gene to gene. **C** AP scores are calculated to evaluate the power of CellProfiler and DeepProfiler features to classify multiple phenotypic classes in Mitocheck morphological data. AP scores capture the ability to retrieve single cells annotated with the same morphological class against negative controls. **D** Mitocheck data, correlation between mAP scores for retrieving single cells annotated with the same morphological class versus gene, for DeepProfiler features. MC: morphological class. mAP *p*-values were estimated using a one-sided permutation test and adjusted for multiple comparisons by Benjamini–Hochberg procedure. Percent retrieved indicates the percentage of scores with *p*-value below 0.05. Source data are provided as a Source Data file.

and consistent perturbations, mAP helps in prioritizing biologically relevant perturbations for deeper mechanistic studies, refining the search space for downstream analyzes such as differential feature identification. Finally, by using metadata-based blocking, mAP assessments can shed light on the effect of technical variation in data (e.g., plate layout effects), suitability of experimental design (e.g., cell type or fluorescent channel selection), and data processing methods (e.g., feature extraction) on phenotypic activity and consistency of profiling data. This adaptability makes mAP a valuable tool for comparing different profiling methods and enhancing the interpretation of high-throughput experiments. Our implementation of the mAP framework in the *copairs* package is highly efficient and scales well to the large-scale datasets, including at the single-cell level (Supplementary Table 2).

At its core, the mAP framework is based on grouping profiles according to the prespecified block design and calculating a well-established evaluation metric on the rank list of nearest neighbors to a given profile. Unlike most existing alternatives[16], this procedure is robust to outliers, fully agnostic to the nature of data, and does not make distributional, linearity, or sample size assumptions. With its top-heavy bias, average precision emphasizes early discovery in ranking assessment similarly to other recently proposed metrics[55–57], but those metrics require careful parameter tuning that can be tricky. Unlike the AP[31], those metrics cannot be interpreted in terms of probability even if they are bounded by [0, 1][56]. If only *k* top ranked perturbations are of interest, requiring the rank list to be thresholded (for example, when the goal is to see how often the correct profile would be in the top k results), AP can be easily replaced by Precision@k.

Still, the mAP framework has limitations. The effectiveness of the mAP framework, like other methods based on nearest neighbors, is contingent upon choosing an appropriate measure of profile dissimilarity (a distance metric). A less-suitable distance metric would impair mAP's performance, but this is a trade-off for the framework's flexibility. Conversely, this opens an opportunity for using a dataset-specific custom similarity measure to suit particular data types and analyses. As a rank-based metric, mAP is robust to deviations from typical assumptions for parametric methods, but it cannot reflect differences in effect size beyond perfect separability between two groups that are being compared. To overcome this limitation, future studies could explore extending mAP to accommodate graded rank lists[32], moving beyond binary classifications. Finally, the permutation testing approach used for significance assessment has limitations when dealing with datasets that have a small number of replicates or controls. This is an inherent statistical constraint that highlights the importance of having adequate experimental replicates and controls for robust statistical analysis.

In conclusion, the mAP framework presents a powerful strategy for evaluating data quality and biological relationships among samples in high-throughput profiling. It adjusts to various data types and perturbations and is robust to the complexities of real-world biological data. It can be effectively used to improve methods and prioritize perturbations for further studies with the potential to streamline the discovery of mechanisms and therapeutic targets in complex biological systems.

## Methods

### mAP calculation

In general, the mAP framework can be used to compare any two groups of high-dimensional profiles by retrieving profiles from one group ("query group") against another group ("reference group"). Groups are defined by providing a list of metadata columns in which profiles that belong to the same group have to have either matching or mismatching values.

Prior to calculating mAP, appropriate preprocessing of profiling data is essential to ensure meaningful similarity comparisons[14,15]. This includes robust standardization to mitigate differences in scale across features, feature selection to reduce noise and focus on informative dimensions, and batch correction to address technical variation across experimental batches. Based on the comparative evaluations across multiple modalities and tasks[17–19], we suggest applying Harmony[58] or Seurat v3[59] before using mAP, as they strike a good balance in preserving meaningful biological structure while mitigating technical variation, making them reliable first choices for general-purpose batch correction.

Given a group of $N$ reference profiles and a group of $M$ query profiles, we calculate non-interpolated AP[29] for each query profile as following:

1. out of $M$ query profiles, select one profile $i$;
2. measure distances from the query profile $i$ to all other $(M-1) + N$ profiles in both groups;
3. rank-order $(M-1) + N$ profiles by increasing distance to the query profile $i$ (decreasing similarity);
4. for each rank $k$ going top-down the list, if $k$ contains another query profile (true positive we term a "correct match", i.e., not a reference), calculate *precision* for this rank $k$;
5. when done, average calculated precisions to obtain the AP value.

More formally, average precision for profile $i$ is calculated as:

$$AP_i = \frac{1}{(M-1)+N} \sum_{k=1}^{(M-1)+N} g_k P_k \qquad (1)$$

where $g_k$ equals 1 if rank $k$ contains a correct match (True Positive) and 0 if otherwise, $P_k = \frac{TP_k}{k}$ is precision at rank $k$ (precision@k), $TP_k$ is the number of all query profiles (all Positives) retrieved up to rank $k$.

More conveniently, AP can be expressed via relative change in recall:

$$AP_i = \sum_{k=1}^{(M-1)+N} (R_{k-1} - R_k)P_k \qquad (2)$$

where $P_k$ is the same as above and $R_k = \frac{TP_k}{M-1}$ (4) is recall at rank $k$ (recall@k), $R_0 = 0$, which replaces both $g_k$ and dividing by $M-1$.

Then, mean AP (mAP) for the whole query group can be calculated through averaging of individual query profile APs:

$$mAP = \frac{1}{M} \sum_{i=1}^{M} AP_i \qquad (3)$$

where $M$ is the number of profiles in the query group.

### Assigning significance to mAP scores

We estimate the statistical significance of a mAP score with respect to a random baseline using a permutation testing approach, a non-parametric, assumption-free method for testing the null hypothesis, which assumes that profiles in both query and reference groups were drawn from the same distribution. Since the total number of points is fixed and the rank list is binary, the mAP distribution under the null hypothesis distribution covering all possible ranking outcomes only depends on two parameters: the number of positives without the query $M-1$ and the total number of points without the query $N + (M-1)$. Therefore, the null has the exact size equal to the binomial coefficient $\frac{N+(M-1)}{M-1}$. In practice, we approximate the null by repeatedly reshuffling the rank list and calculating mAP, which is equivalent to reshuffling the profile labels. The $p$-value is then calculated as the fraction of the approximate null that is greater than or equal to the mAP score. This approach aligns with the interpretation of significance values in parametric statistical analyzes, where a nominal significance cutoff of 0.05 is typically used. When we compare mAP scores of multiple query groups, we correct corresponding $p$-values for multiple comparisons using the Benjamini–Hochberg procedure[40]. We refer to the percentage of calculated mAP scores with a corrected $p$-value below 0.05 as the *percent retrieved*.

### mAP for phenotypic activity and consistency assessment

We applied the mAP framework to assess phenotypic activity and consistency.

We assess phenotypic activity of a single perturbation by calculating mAP for replicate retrievability, i.e., the ability to retrieve a group of perturbation's replicates (query group) against a group of control profiles (reference group). At this stage, a replicate profile typically means an aggregation of single-cell profiles (e.g., across all cells in a single well). By imposing additional conditions, we defined various groups of replicates for a given perturbation. For example, we used phenotypic activity to evaluate the presence of plate effects by comparing mAP score for retrieving replicates from the same plate vs from different plates. After calculating mAP scores for all perturbations, they can be compared and ranked in terms of their phenotypic activity.

We also use mAP to assess the phenotypic consistency of multiple perturbations annotated with common biological mechanisms or modes of action (query group) against perturbations with different annotations (reference). When computing phenotypic consistency, each perturbation's replicate profiles are first aggregated into a consensus profile by taking the median of each feature to reduce profile noise and improve computational efficiency.

Let's consider a dataset containing perturbations annotated with mechanisms of action. For example, a group of $P$ compounds is annotated with $MoA_1$, and the rest $Q$ compounds are annotated with various other $MoA$ labels.

Then the mAP for the $MoA_1$ group of $P$ perturbations can be computed as following:

1. select one perturbation profile from this group, e.g., $P_i$;
2. measure distances from $P_i$ to all other $(P-1)+Q$ profiles in both groups;
3. rank-order $(P-1)+Q$ profiles by decreasing similarity w.r.t to $P_i$
4. going top-down the list, if the rank $k$ contains a perturbation profile from the same group $P$, calculate *precision@k* for this rank $k$
5. when done, average calculated precisions by summing them up and dividing by $P-1$
6. repeat the process for all $i=1...P$ and average obtained APs to calculate $mAP_P$

The resulting value $mAP_P$ will indicate how internally consistent (has high mAP for retrieving perturbations from itself) this group of perturbations annotated with $MoA_1$ is compared to other perturbations. This example can be easily extended to an arbitrary number of perturbation groups (e.g., compound MoAs). The same process can also be repeated using each set of perturbations as a query group. This will result in obtaining mAP scores for all groups of perturbations in the dataset and can be used to rank them by phenotypic consistency or estimate the consistency of the whole dataset by aggregating them (e.g., by averaging).

Additionally, we can also define *phenotypic distinctiveness*, although it is not used in this paper. While phenotypic activity measures how distinguishable a perturbation is from negative controls, phenotypic distinctiveness measures how distinguishable a perturbation is from all other perturbations in the experiment. It can be assessed by calculating mAP for retrieving the replicates of a perturbation against all other perturbations. This concept is essentially the same as the "mAP-nonrep" score used in ref. [19].

### Extension to multiple labels

When considering groups of perturbations, a single perturbation can belong to multiple groups simultaneously. For example, a compound can have multiple annotations, such as genes whose products are targeted by the compound, or mechanisms of action of this compound. Then AP can be calculated by considering a single annotation group at a time. In the example below, we assume having per-perturbation aggregated consensus profiles.

Let's consider a dataset containing consensus profiles of $P$ perturbations, with each perturbation annotated with "labels" from $1...T$, where $T$ is the number of all possible labels in the dataset.

Then for a label and one of the perturbations annotated with it, AP can be calculated as:

1. select one label $t$ from $T$ possible options
2. select one perturbation profile $p_t$ out of $P_t$ perturbations annotated with this label (query)
3. rank-order the rest of profiles $(P-1)+1$ by similarity w.r.t to $p_t$
4. going top-down the list, if the rank $k$ contains a perturbation profile that is also annotated with the label $t$, calculate *precision@k* for this rank $k$
5. when done, average calculated precisions by summing them up and dividing by $P_t$, i.e., the number of all perturbations annotated with this label
6. the result will be AP for the specific $t$-$p_t$ label-perturbation pair
7. repeat steps 2-6 for all perturbation profiles $P_t$ to obtain APs for all perturbations annotated with this label $t$
8. repeat steps 1-7 for all labels $T$ to obtain APs for all label-perturbation pairs

The result will be a sparse $P \times T$ matrix of APs, where the element corresponding to a perturbation $p$ and target $t$ is equal to $AP_{t\text{-}p}$ if $p$ is annotated with $t$ and 0 otherwise. This matrix can be aggregated on a per-perturbation or per-label basis (for example, by taking the mean across rows or columns, correspondingly) depending on the downstream task. Per-label mAP will assess biological consistency of perturbations annotated with a specific label compared with perturbations annotated with other labels. Practically, this makes it possible to compare consistency of different label groupings for a given perturbation.

### Simulated data generation protocol

Simulations of the mAP performance were conducted by repeatedly generating control and treatment replicates by sampling features from a number of different normal distributions. Each treatment was simulated in 2,3 or 4 replicates, and 8, 16, or 32 replicates were simulated for each control. Between 100 and 5000 features were simulated. All features were simulated in the control by sampling from the standard normal distribution. Varying numbers of features were simulated in treatment replicates by sampling from a shifted normal distribution ($\mu = 1$, $\sigma = 1$). Any remaining features in treatment replicates were sampled from the standard normal distribution. Each perturbation was considered correctly retrieved if its $p$-value was below 0.05.

### Alternative metrics

The multidimensional perturbation value (mp-value)[22] is a statistical metric designed to assess differences between treatments in various types of multidimensional screening data. It involves using principal component analysis (PCA) to transform the data, followed by calculating the Mahalanobis distance between treatment groups in this PCA-adjusted space. The significance of the mp-value is determined through permutation tests, a non-parametric approach that reshuffles replicate labels to assess the likelihood of observed differences occurring by chance.

The Maximum Mean Discrepancy (MMD)[25] test is a multivariate nonparametric statistical test used to determine if two distributions are significantly different. It measures the largest possible difference in expectations across a function space, typically within a reproducing kernel Hilbert space (RKHS). We use MMD with the radial basis function kernel (RBF) and set the kernel bandwidth at the median distance between points in the aggregate sample, a common heuristic[25].

The k-means algorithm clusters data by minimizing within-cluster variance, effectively grouping samples based on their similarity. We set the number of groups $k$ to 2 for separating perturbation and control replicates. Cluster centroids are initialized randomly, and the algorithm is repeated 10 times, with the best result selected based on the lowest overall inertia (the sum of squared distances of samples to their closest cluster center), as implemented in scikit-learn[60].

### Cell Health dataset description and preprocessing

We used our previously published "Cell Health" dataset[42] of Cell Painting[27] images of CRISPR-Cas9 knockout perturbations of 59 genes, targeted by 119 guides in three different cell lines (A549, ES2, and HCC44). Morphological profiles were previously extracted from images using CellProfiler[52] and median-aggregated on the well level[42]. We used a subset of 100 guides that had exactly six replicates (two replicates in three different plates) in each cell line. We performed two types of profile preprocessing followed by feature selection using pycytominer[39]. The first preprocessing method included data standardization by subtracting means from feature values and dividing them by variance using the whole dataset. Alternatively, we used a robust version of standardization, which replaces mean and variance with median and median absolute deviation, correspondingly, and is applied on a per-plate basis ("MAD robustize"). Feature selection

included variance thresholding to remove features with minimal variation across the dataset, removing highly correlated features, removing features with missing values or outliers, and removing blocklisted features—all using pycytominer[39] default parameters.

## cpg0004 dataset description and preprocessing

We used our previously published dataset "cpg0004-lincs" (abbreviated to cpg0004 here) that contains Cell Painting[27] images of 1,327 small-molecule perturbations of A549 human cells[11]. The wells on each plate were perturbed with 56 different compounds in six different doses. Every compound was replicated 4 times per dose, with each replicated on a different plate. In this study, only the highest dose point of 10 µM was used. Morphological profiles were previously extracted from images using CellProfiler[52]. Profile preprocessing, feature selection, and batch correction were performed using pycytominer[39]. First, profiles were re-scaled against DMSO controls by subtracting medians from DMSO feature values and dividing them by median absolute deviation ("MAD robustize"). Feature selection included variance thresholding to remove features with minimal variation across the dataset, removing highly correlated features, and removing features with missing values or outliers. Finally, profiles were corrected for batch effects by the sphering transformation[19] (computes a whitening transformation matrix based on negative controls and applies this transformation to the entire dataset).

## cpg0016[orf] dataset description and preprocessing

We used the JUMP Consortium's[13] "cpg0016-jump[orf]" dataset[46] (abbreviated to "cpg0016[orf]" here), which contains Cell Painting[27] images of U2OS cells treated with 15,136 overexpression reagents (ORFs) encompassing 12,602 unique genes. Morphological profiles were previously extracted from images using CellProfiler[52], mean-aggregated on the well level, and then corrected for plate layout effects by subtracting means from feature values per well location. Cell counts were regressed out from each feature with more than 100 unique values. After that, profiles were preprocessed per plate by subtracting medians from feature values and dividing them by median absolute deviation ("MAD robustize"). Feature selection was performed using pycytominer[39] and profiles were corrected for batch effects by a combination[19] of the sphering transformation and Harmony[58] (an iterative algorithm based on expectation-maximization that alternates between finding clusters with high diversity of batches, and computing mixture-based corrections within such clusters).

## nELISA dataset description and preprocessing

We used the dataset containing proteomic profiles from a 191-plex nELISA[12], a high-throughput, high-plex assay designed for quantitative profiling of the secretome, which was performed in A549 cells across 306 well-characterized compound perturbations from the Broad Institute's drug repurposing library[49]. This dataset also included matching CellProfiler[52] morphological profiles from Cell Painting[27] images of the same physical samples whose supernatants were nELISA-profiled. Profiles were preprocessed per-plate by subtracting medians from feature values and dividing them by median absolute deviation ("MAD robustize"). Feature selection included variance thresholding to remove features with minimal variation across the dataset, removing highly correlated features, and removing features with missing values or outliers.

## Perturb-seq dataset description and preprocessing

We used the public Perturb-seq[7–10] mRNA profiling dataset of single cells treated with CRISPRi containing 10X single-cell gene expression reads, barcode identities, and activity readouts (Gene Expression Omnibus accession GSE132080)[61]. The experiment assessed how single-guide RNAs (sgRNAs) containing mismatches to their target

sites attenuate expression levels of target genes[50]. Specifically, 25 genes involved in a diverse range of essential cell biological processes were targeted with 5–6 mismatched sgRNAs, covering the range from full to low activity, and 10 nontargeting controls. Each mismatched guide was characterized by its activity levels relative to the perfectly matched sgRNA targeting the same gene[50]. The distributions of sgRNAs were largely unimodal, although broader than those with the perfectly matched sgRNA or the control sgRNA[50]. We performed single-cell profile normalization and feature selection using Seurat[59].

## Mitocheck data description and preprocessing

We used the previously published Mitocheck dataset[51] containing images of GFP-tagged nuclei of HeLa cells perturbed with small interfering RNA (siRNAs) to silence approximately 21,000 protein-coding genes. Within the dataset, approximately 3000 cell images were manually labeled into one of 15 morphological phenotype classes. Recently, these images were re-analyzed[54] with a more comprehensive image analysis pipeline, which included illumination correction using PyBasic[62], segmentation using CellPose[63], and single-cell feature extraction using CellProfiler[52] and DeepProfiler[53]. Extracted profiles were standardized by removing the mean and scaling to the unit variance of negative control cells. We performed feature selection for both CellProfiler- and DeepProfiler-derived profiles by variance thresholding to remove features with minimal variation across the dataset, removing highly correlated features, removing features with missing values or outliers, and removing blocklisted features—all using pycytominer[39] default parameters.

## Reporting summary

Further information on research design is available in the Nature Portfolio Reporting Summary linked to this article.

## Data availability

Profiles extracted from the Cell Health dataset[42] are available at https://github.com/broadinstitute/cell-health/tree/30ea5de393eb9c fc10b575582aa9f0f857b44c59/1.generate-profiles. Profiles extracted from the cpg0004[11] dataset are available at https://github.com/broadinstitute/lincs-cell-painting/tree/061870127481dcd73c29df85e bcfddeac2ed0828/profiles. Profiles extracted from the cpg0016[orf][46] dataset are available from the Cell Painting Gallery[13] at https://github.com/broadinstitute/cellpainting-gallery/blob/87e046 96564e8c61d060c2a8e3a99dbd00fd9b31/README.md. The Perturb-seq dataset is available at Gene Expression Omnibus, accession code GSE132080[61] [https://www.ncbi.nlm.nih.gov/geo/query/acc.cgi?acc= GSE132080]. Profiles extracted from the matching nELISA-Cell Painting dataset[12] are available at https://github.com/carpenter-singh-lab/ 2024_Kalinin_mAP/tree/e9a5414726119dca7ed0d79efde887c1e259 c288/experiments/5_nelisa/inputs. Profiles extracted from the Mitocheck dataset[51] are available at https://github.com/WayScience/ mitocheck_data/blob/613acbb20d2134ad1d725c7605a61c5a9e823c1a/ README.md. Source data are provided with this paper.

## Code availability

The mAP framework is implemented as an open-source Python package *copairs*, available at https://github.com/cytomining/copairs under the BSD 3-Clause license. The implementation relies on numpy[64], scipy[65], pandas[66], tqdm[67], duckdb[68], and statsmodels[69] open-source Python packages. The code for downloading data, performing analyzes, and generating results in this study is publicly available and has been deposited in Gihub at https://github.com/carpenter-singh-lab/ 2025_Kalinin_mAP under the BSD 3-Clause license. The specific version of the code associated with this publication is archived in Zenodo and is accessible via https://doi.org/10.5281/zenodo.15151267[70].

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

## Acknowledgements

We thank members of the labs of JT Neal, Paul Blainey, Frank Li, and Calvin Jan for their input on ideas that led to the work presented in this paper. We thank Erin Weisbart, Nodar Gogoberidze, Ellen Su, Srinivas Niranj Chandrasekaran, and Jessica Ewald for the feedback provided to improve the clarity of this manuscript. We also would like to thank Jenna Tomkinson and Roshan Kern for MitoCheck data pre-processing. This work was partially supported by Calico Life Sciences, LLC, the Human Frontier Science Program (RGY0081/2019 to Sh.S.), and a grant from the National Institutes of Health NIGMS (R35 GM122547 to A.E.C.). Research reported in this publication was supported in part by the National Library of Medicine (NLM) of the National Institutes of Health (NIH) under award number T15LM009451 to E.S. G.P.W. and E.S. were supported by an American Heart Association Collaborative Sciences Award (24CSA1255857) to G.P.W. This research was, in part, funded by the United States Government, ARPA-H (grant number 1AY2AX000005-01 to Johan Paulsson). The views and conclusions contained in this document are those of the authors and should not be interpreted as representing the official policies, either expressed or implied, of the United States Government.

## Author contributions

A.A.K., A.E.C., G.P.W., and Sh.S. conceived the study. A.A.K., J.A., B.R., G.P.W., and Sh.S. designed the mAP framework. A.A.K., J.A., and A.F.M. implemented the copairs package. A.A.K., J.A., E.S., Su.S., and A.F.M. tested and documented the copairs package. A.A.K., J.A., L.V., E.S., H.T., M.B., A.F.M., and G.P.W. performed data processing and analysis. A.A.K., A.E.C., G.P.W., and Sh.S. wrote and edited the manuscript with input from other authors. All authors approved the final version of the manuscript.

## Competing interests

The Authors declare the following competing interests: Sh.S. and A.E.C. serve as scientific advisors for companies that use image-based profiling and Cell Painting (A.E.C.: Recursion, SyzOnc, Quiver Bioscience, Sh.S.: Waypoint Bio, Dewpoint Therapeutics, Deepcell) and receive honoraria for occasional scientific visits to pharmaceutical and biotechnology companies. All other authors declare no competing interests.
