## [Transparent Peer Review file · Nature Communications]

A versatile information retrieval framework for evaluating profile strength and similarity

Corresponding Author: Dr Shantanu Singh

Version 0:

Reviewer comments:

Reviewer #1

(Remarks to the Author)

Kalinin et al. introduce a statistical framework that uses mean average precision as a data metric to separate biological signal and technical noise. They demonstrate the use of mAP on several tasks including perturbation scoring across image, mRNA, and protein profiles and different perturbation types. Overall the paper is well written, but I have a few suggestions that will improve the overall message of the paper.

There are two things that a researcher is often interested in: 1) features that show replicability and 2) strong SNR. The key idea behind mAP is it can retrieve other samples from the same group (1) from a collection that also includes samples from other group. To achieve (2), the researchers using mAP have to compare it against multiple contrasts. But in its current form, the text does not do a good job of demonstrating that (2) is achieved (this for context is what is represented in Figure 1F by phenotypic consistency).

- Figure 1 is very nice and lays down the objective and approach of the paper. For Figure 2, while choosing the proportion of features, the authors chose from the range 1-64%. While I can see what the goal of this exercise is, I think a more sensible choice of feature set would be more regular. 1-100%. The x-axis on Figure 2 is a bit unintuitive to read. Also, I cannot think of a theoretical reason the simulation should stop at 64% different features - this number could be large in certain single-cell contexts.

- Extending Figure 3D, can the authors also show the phenotypic consistency against other genes when all three cell lines are considered together - the goal here would be to see which genes have reproducible phenotypes (across cell types). The authors can also choose to show the impact of each gene (on a scatter plot) similar to their single-cell analysis, but here for celltypes. One thing that is currently not clear is which genes reproduce across all the cell types (and what impact these are having)

- For Figures 4 and 5, what does the distribution of guide mismatches look like? Is it unimodal with 1-2 mismatches or is it more uniform with 1-5 mismatches?

- It would be great if the authors could show some downstream impact of using the strongest phenotypic consistent sets. For example, does the phenotypic consistent feature set also result in "tighter" clustering? I think that would complement Figure 5C which tries to show the "physiological impact" of the consistent set

Minor comments

- Figure 2 - the $\exp(2)$ scaling on the X axis is confusing, please consider using full scope (1-100)
- Line 549-550 Precision@k is never really defined

Software comments

- I was able to install and run the example given on the software's github
- However, it would be wonderful if the authors could provide more detailed vignettes - this would ensure end users can easily integrate mAP scoring in their workflows

(Remarks on code availability)

Software comments

- I was able to install and run the example given on the software's github
- However, it would be wonderful if the authors could provide more detailed vignettes - this would ensure end users can easily integrate mAP scoring in their workflows

Reviewer #2

(Remarks to the Author)

The paper by Kalinin et al. proposed a rank-based single-variable metric mAP (i.e. mean average precision) to evaluate the similarity across multiple biological samples (e.g. profiles of images, protein, mRNA, CRISPR or small molecular perturbations, and single-cell experiments). By quantitatively evaluate the replicability of specific types of profiles (i.e. replicate retriability), the paper suggests that mAP is a reasonable metric to delineate the phenotype activities of specific factors and phenotype consistency among profiles associated with the same perturbations, against technical noise or variations.

While robust quantification of biological differences across multiple profiles is a common problem in data analysis, the novelty of the proposed method and its advantages over other approaches are not clearly justified. It is also not clear what new biological insights can be obtained using mAP, since its calculation largely depends on pre-existing labels of biological profiles.

Major comments:

1. Although the authors claim that mAP is not dependent on specific choices of distances between profiles, there is no evidence provided in the current manuscript. Considering the complex nature of high-dimensional biological experiments, the authors need to benchmark the performance of mAP based on different types of distances.

2. Each biological profile is usually mixed with a combination of multiple biological variations and technical factors, which typically requires joint modeling of the variations/factors together. However, mAP requires the users to pre-select one type of variations/factors to evaluate the profile similarities, based on pre-existing profile labels or meta-data.

For example, in the manuscript, the authors usually specify one factor at a time for comparisons or group profiles with similar MoA together. Does that mean that mAP can only leverage the marginal distribution information, instead of the joint distribution information?

If this is true, then the practical use of mPA seems to be limited as a feature selection approach.

3. In practice, many genome-wide biological profiles only differ from each other in a small subset of regions/genes. However, in Fig 2, it seems the performance of mAP is usually lower than mp-value, for small percentages of differential features. Does that mean that mAP can not efficiently dissect phenotypic activities when the number of differential features (e.g. differential expressed genes) are relatively small?

4. Related to the comment above, beyond the basic evaluation of global similarities/differences across profiles, it is usually more desired to pinpoint the specific differential features in biological studies. Could mAP also prioritize such features?

5. The permutation test for p-value calculation should be based on re-shuffling of profile labels, instead of just shuffling the ranked lists.

6. The simulated datasets are based on normal distributions, which is very rare in real biological profiles. The authors need to benchmark the performance using additional distributions.

Also, the authors also need to test the performance based on different variances. Currently, the variances are fixed to be the same.

7. In Fig 3A, "Results are shown for a subset of an arbitrary pair of plates as a proof-of-concept". The authors need to show

the complete set of results (maybe as supplementary figures if needed).

8. The authors may compare mAP versus a basic approach of directly clustering the profiles based on the multi-variable features and check how separable the profiles are clustered.

9. As a suggestion, I wonder whether the authors could demonstrate the advantage of mAP in the context of data sparsity, which is a common problem in single-cell profiles. If mAP can efficiently delineate the profiles even if the profiles contain highly sparse features, that would strengthen the significance.

10. In lines 439-443: the 4% retrieval for nELISA seems very low. The authors may further clarify whether it is due to the limitation of mAP.

11. In lines 389-393: the authors briefly mentioned the issues of incomplete and imperfect profile annotations. Since mAP requires the pre-existing profile annotations/labels, it seems to be an important problem. The authors may further discuss this issue and how to overcome in practical implementations.

(Remarks on code availability)

Overall the GitHub repository is well organized with clear structure and readme for installation and usage.

The authors may need to provide additional notations for the scripts in "tests".

Reviewer #3

(Remarks to the Author)

The manuscript proposes a framework based on the mean average precision (mAP) to analyze and compare high-dimensional profiles. As the authors acknowledge, mAP is a routinely used metric in machine learning, and the proposed permutation test is also standard in statistics. The proposed framework appears to be a direct application of mAP with limited novelty. Another major concern is that it's difficult to justify the use of this framework in real practice given the existence of many easier alternatives.

1. Consider a typical analysis involving two high-dimensional profiles. It is difficult to envision a scenario where the focus would be on testing between multivariate data. In most biomedical and biological applications, including those discussed in the manuscript, marginal testing of individual features is more useful. For example, in the perturbation and single-cell data analysis, why would one choose mAP over more straightforward and informative test of individual genes?

2. In the application to the perturbation data, the manuscript demonstrates that the mAP can be used assess batch effects. However, the manuscript does not discuss why this method is better than conventional methods, for example, using a clustering analysis after dimensionality reduction or a direct differential test with respect to the batch effect. Both methods are much more straightforward and easier to interpret than the propose mAP.

3. The mAP framework relies on permutation to obtain statistical significance. For each set of permuted data, a pairwise distance matrix needs to be calculated. Such an approach will be computationally intensive when sample size is large. It may not be feasible in single-cell studies with tens of thousands of cells.

4. There is a rich body of literature in statistics discussing multivariate tests, such as Hotelling's test and kernel tests. However, there is a lack of proper discussion and comparison with these methods.

5. In the comparison between CellProfiler and DeepProfiler data, the authors reported varying mAP values and retrieval rates. How to evaluate the performance of mAP from these values? What mAP value is considered large or small?

6. The definition of the retrieval rate is unclear.

(Remarks on code availability)

The software is made available. However, the code and data for reproducing the analysis are not available.

Version 1:

Reviewer comments:

Reviewer #1

(Remarks to the Author)

The authors have addressed all my comments. New Figure 3 is great. No further suggestions from my side.

(Remarks on code availability)

Addition of new jupyter notebooks is great and will increase the usability of this package.

Reviewer #2

(Remarks to the Author)

The authors have addressed the majority of my original comments. For the remaining comments (mostly regarding the overall significance and innovation of this study), the authors mainly added texts in the manuscript to admit the limitations (such as the ability of prioritizing specific differential features, overcoming the sparsity issues of input data in practice, and performing robustly based on different choices of distance metrics).

Although my enthusiasm on this paper is dampened due to the limited methodology innovation, an easy-to-use tool, as presented in the revised manuscript and their GitHub, may be valuable for practical applications of multiple communities.

(Remarks on code availability)

The GitHub repository is well organized and annotated for use.

Reviewer #3

(Remarks to the Author)

The revised manuscript has addressed some of the questions I previously raised; however, two major concerns remain inadequately addressed.

1. My first major concern relates to the advantage of mAP over methods designed to identify individual differential features, such as genes. Reviewer 2 raised a similar question, and while the authors' response provides some clarification, the utility of mAP in this context remains unclear. It is true that comparing high-dimensional data and identifying individual differential features are distinct computational problems. However, the manuscript fails to present a compelling argument as to why mAP analysis would become a widely adopted upstream analysis for differential feature analysis.

An important question to address is what additional biological insights can be gained from conducting mAP analysis, especially if it cannot replace feature-based analysis. Clarifying this point is crucial for demonstrating the broad applicability of the method.

2. My second concern pertains to mAP's utility for evaluating batch effects compared to existing, more straightforward methods. The authors referred to their recent preprint on bioRxiv, arguing that "no single metric could fully evaluate batch effect correction." While this may be true, it does not establish that mAP is generally a better tool for assessing batch effects.

First, the preprint referenced primarily focuses on image-based cell profiling, as suggested by its title, "Evaluating batch correction methods for image-based cell profiling." This restricts its scope and does not address batch effect correction across diverse data types. Second, there does not appear to be any direct comparison between mAP and existing batch effect evaluation methods in the preprint's main figures. Third, the authors mentioned Supplementary Figure 7 in the preprint as containing such a comparison, but the supplementary file is unavailable on its bioRxiv page.

Rather than relying on a reference to this preprint, which has a different focus, it would be far more convincing to include a direct comparison between mAP and other established batch effect evaluation methods within this manuscript. This would provide readers with clearer evidence of mAP's utility and advantages.

3. The authors acknowledged that computational time on large datasets could be a concern, and they have improved the efficiency of the code. It would be helpful to report the current computational time needed for each dataset in the manuscript to give readers a rough sense. A discussion of the theoretical computational complexity will be even better.

(Remarks on code availability)

Previous comment was addressed by adding example scripts.

Version 2:

Reviewer comments:

Reviewer #3

(Remarks to the Author)

The authors' new responses have resolved my questions about the differential feature analysis and computational time. Regarding the batch effects, the authors clarified that mAP is not specifically designed for assessing batch effects. It would be helpful to clarify in the manuscript what the suggested pipeline is when batch effects are expected to be present in the data. Should users apply another batch effect removal method first before applying mAP, or is there a way to first evaluate whether mAP can handle the batch effects in the specific case?

(Remarks on code availability)

Dear reviewers:

We are thankful for your time and efforts in providing us with constructive and encouraging feedback. We have carefully reviewed the comments and revised the manuscript accordingly. Corresponding changes to the manuscript text are highlighted in yellow. Numerical references in this letter correspond to citation numbers in the body of the revised manuscript.

Our point-by-point responses are given below, but we'd like to highlight other substantial changes to the manuscript and our software that involved reprocessing some analyses:

- We've streamlined the approach to calculating mAP p-values. Due to changes in the implementation, for some datasets we calculated a p-value for each AP score, performed FDR correction, and then combined AP p-values into a mAP p-value using Fisher's method in the previous version of the manuscript. However, this approach is indirect and Fisher's method may incorrectly estimate the combined p -value, especially for a rank-based statistic. Because our test statistic of interest is mAP itself, a more straightforward approach is to compute a p-value directly for mAP. We've updated our analyses to make sure we use direct mAP p-value calculation for obtaining all results in the paper.
- We've updated the protocol for running simulations used to make Figure 2. Specifically, in the previous version each time the dataset was regenerated from a different random seed. In the revised version, the same random seed is used to generate each dataset, leading to more consistent performance along the X axis as more features differ from controls.

Reviewer #1

Kalinin et al. introduce a statistical framework that uses mean average precision as a data metric to separate biological signal and technical noise. They demonstrate the use of mAP on several tasks including perturbation scoring across image, mRNA, and protein profiles and different perturbation types. Overall the paper is well written, but I have a few suggestions that will improve the overall message of the paper.

R1.0 There are two things that a researcher is often interested in: 1) features that show replicability and 2) strong SNR. The key idea behind mAP is it can retrieve other samples from the same group (1) from a collection that also includes samples from other group. To achieve (2), the researchers using mAP have to compare it against multiple contrasts. But in its current form, the text does could be stronger in demonstrating that (2) is achieved (this for context is what is represented in Figure 1F by phenotypic consistency).

We thank the reviewer for constructive suggestions to improve the manuscript. We followed the reviewer's specific feedback and addressed this under queries **R1.2** and **R1.4** below.

R1.1 Figure 1 is very nice and lays down the objective and approach of the paper. For Figure 2, while choosing the proportion of features, the authors chose from the range 1-64%. While I can see what the goal of this exercise is, I think a more sensible choice of feature set would be more regular. 1-100%. The x-axis on Figure 2 is a bit unintuitive to read. Also, I cannot think of a theoretical reason the simulation should stop at 64% different features - this number could be large in certain single-cell contexts.

Thank you for pointing out that the x-axis may be unintuitive to read. The purpose for binary exponential scaling on the x-axis was to demonstrate the range from a very few features differing (i.e. for a profile with 100 features, 1% is a just single feature different from controls, 2% is two features, etc) to more than a half differing (64%). In doing so, we followed the mp-value¹⁴ article that introduced this type of scale. To demonstrate results across the whole 1-100% range, we extended evaluations performed in Figure 2 and added new Supplementary Figure S2 showing these results. We also refer to this new figure in the main text (line 283).

R1.2 Extending Figure 3D, can the authors also show the phenotypic consistency against other genes when all three cell lines are considered together - the goal here would be to see which genes have reproducible phenotypes (across cell types). The authors can also choose to show the impact of each gene (on a scatter plot) similar to their single-cell analysis, but here for celltypes. One thing that is currently not clear is which guides reproduce across all the cell types (and what impact these are having)

Neat idea! We reworked the whole Figure 3 to show three cell lines together in all analyses. We also added color-coded cell line-specific retrieval rates and new panels containing mosaic scatter plots to show the impact and agreement across cell lines both for guide (Figure 3C) and gene (Figure 3F) retrieval. We updated the description of these results in the main text (lines 335 through 416).

R1.3 For Figures 4 and 5, what does the distribution of guide mismatches look like? Is it unimodal with 1-2 mismatches or is it more uniform with 1-5 mismatches?

The original study⁴⁵ that produced the Perturb-seq dataset showed that the distributions of sgRNAs were largely unimodal, although broader than those with the perfectly matched sgRNA or the control sgRNA. We added this information to the Perturb-seq dataset description in Methods (lines 858-860).

Additionally, we also added a new Supplemental Figure S8, demonstrating how single-cell AP scores reflect impact of individual guides.

R1.4 It would be great if the authors could show some downstream impact of using the strongest phenotypic consistent sets. For example, does the phenotypic consistent feature set also result in "tighter" clustering? I think that would complement Figure 5C which tries to show the the "physiological impact" of the consistent set

This is a nice suggestion to showcase that better retrieval rates can be reflected in tighter clustering. We added a new Supplementary Figure S10 to visualize UMAP embeddings of single-cell profiles of four exemplar morphological classes (a subset from Figure 5C, as suggested) with varying retrieval rates. As expected, clusters were more consistent for morphological classes with higher retrieval rates. We referred to this result in the main text (lines 564-566).

R1.4 Minor comments

- **Figure 2 - the exp(2) scaling on the X axis is confusing, please consider using full scope (1-100)**
- **Line 549-550 Precision@k is never really defined**

We updated Figure 2 and added new Supplementary Figure S2 as discussed in response to **R1.1**. We also added an explicit definition of Precision@k (lines 181, 658).

R1.5 Software comments

- **I was able to install and run the example given on the software's github**
- **However, it would be wonderful if the authors could provide more detailed vignettes - this would ensure end users can easily integrate mAP scoring in their workflows**

We added example Jupyter notebooks to the package repository demonstrating how to group profiles based on their metadata and how to calculate mAP for both phenotypic activity and consistency assessment (<https://github.com/cytomining/copairs/tree/main/examples>).

Reviewer #2:

The paper by Kalinin et al. proposed a rank-based single-variable metric mAP (i.e. mean average precision) to evaluate the similarity across multiple biological samples (e.g. profiles of images, protein, mRNA, CRISPR or small molecular perturbations, and single-cell experiments). By quantitatively evaluate the replicability of specific types of profiles (i.e. replicate retriability), the paper suggests that mAP is a reasonable metric to delineate the phenotype activities of specific factors and phenotype consistency among profiles associated with the same perturbations, against technical noise or variations.

R2.0 While robust quantification of biological differences across multiple profiles is a common problem in data analysis, the novelty of the proposed method and its

advantages over other approaches are not clearly justified. It is also not clear what new biological insights can be obtained using mAP, since its calculation largely depends on pre-existing labels of biological profiles.

We thank the reviewer for the detailed and insightful feedback.

One key aspect we aimed to highlight is that evaluating profiles has not been commonly approached as an information retrieval problem. In this paper, we offer a new perspective on this problem rooted in our thinking about the properties of the data (e.g., high throughput with few replicates and many features that are often non-Gaussian) and the potential use cases (for example, comparing a profile of a previously unseen perturbation with existing datasets to characterize its strength and/or similarity to known modes of action). We demonstrated the advantages of the proposed approach over established techniques by emphasizing the theoretical properties of mAP (Introduction, lines 93-110; Discussion, lines 590-615), and by benchmarking its performance in controlled scenarios using simulated data that were extended at the request of reviewers (updated Figure 2, new Supplementary Figures S2-S5). We further demonstrated utility of mAP across various profile types (image, protein, and mRNA), perturbation types (CRISPR gene editing, gene overexpression, and small molecules), resolutions (single-cell and bulk), and tasks (assessing technical variation, phenotypic activity of perturbations, and consistency of their groupings)—all within a single unified framework. Moreover, we showed that mAP is useful to evaluate the quality of profiling data using information about the experiment itself (conditions, samples, batches, etc), without the need for external labels (updated Figure 3A). While one would not expect an improved metric, or evaluations thereof, to itself yield biological discoveries, we do see clear paths for this approach to provide useful insights in the design of an experiment and demonstrate potential for biological applications. For example, mAP can guide the selection of alternative experimental variables, such as timepoints or staining conditions, in order to optimize the retrieval of samples expected to look alike. As another example, if using mAP shows that perturbations used in the experiment are not phenotypically active, they might be not relevant to the chosen cell type or the kind of readout that this experiment produced, see **R2.10**).

Finally, we hope to make the mAP framework as easy to use as alternatives through the release of our open-source library that implements: (i) a flexible framework for profile grouping using metadata-based block design, and (ii) mAP calculation optimized processing large datasets (see **R3.3**). We added demo examples showing applications to real data (see **R1.5**), and a separate repository with full source code for reproducing all results in this manuscript (see **R3.7**).

We edited the Introduction (lines 93-110) and Discussion (lines 590-604) to further emphasize the novelty and adaptability of our approach. To further demonstrate the advantages of mAP we added more simulated scenarios to comparisons with existing alternatives (Figure 2, Supplementary Figures S2-S5).

Major comments:

R2.1 Although the authors claim that mAP is not dependent on specific choices of distances between profiles, there is no evidence provided in the current manuscript. Considering the complex nature of high-dimensional biological experiments, the authors need to benchmark the performance of mAP based on different types of distances.

We thank the reviewer for pointing out that the importance of choosing an appropriate distance could be stated more explicitly and illustrated by an example. We added benchmarking the performance of mAP using different types of distances on simulated data (Supplementary Figure S5). While the choice of distance metric affected the results, the recall performance of mAP remained competitive with existing methods. However, in practice this remains a hyperparameter that needs to be chosen based on the nature of the data at hand. To clarify this, we edited the main text to emphasize the importance of choosing the appropriate distance metric for a given data type (Results, lines 195-204). We also highlighted this issue more prominently in the Discussion section by describing the framework's limitations with respect to this important consideration (lines 617-622).

R2.2 Each biological profile is usually mixed with a combination of multiple biological variations and technical factors, which typically requires joint modeling of the variations/factors together. However, mAP requires the users to pre-select one type of variations/factors to evaluate the profile similarities, based on pre-existing profile labels or meta-data.

For example, in the manuscript, the authors usually specify one factor at a time for comparisons or group profiles with similar MoA together. Does that mean that mAP can only leverage the marginal distribution information, instead of the joint distribution information?

If this is true, then the practical use of mPA seems to be limited as a feature selection approach.

Our framework provides an ability to group profiles in structured block designs that incorporate one or more metadata variables. An example of the simplest design would be having a single perturbation with multiple replicates and multiple control replicates (Figure 1A without perturbation 2). In that case all perturbation replicates form one group and all control replicates form another one (Figure 1B), without using any other metadata information that these profiles might have (batch, plate, well position, etc.). Then, the calculated mAP score will account for all sources of variability in and between these groups, i.e. use joint distribution information that incorporates multiple biological and technical factors simultaneously. However, our framework also allows for reducing unexplained variability by using block designs, in which additional metadata variables are used to specify how groups of profiles are formed. Examples of this are given in Figure 3:

- Batch and well position metadata is used to disentangle their effect on phenotypic activity by analyzing guide replicates across plate/well combinations (Figure 3A);
- Fluorescence channel metadata are used to isolate contribution of different organelles on phenotypic activity (Figure 3D);
- Cell line information is used to disentangle biological variability in phenotypic activity (Figures 3A-D) and phenotypic consistency (Figures 3E-F).

These analyses illustrate that the proposed framework can be used in both ways: jointly accounting for all sources of variability or using available metadata to minimize the impact of their variability on the observed scores.

We edited the manuscript to explicitly mention how metadata-based block designs can be used to define profile groupings (Results line 123-131, 139-147). We also included an example of grouping profiles based on predefined metadata rules with our software package:

https://github.com/cytomining/copairs/blob/main/examples/finding_pairs.ipynb.

R2.3. In practice, many genome-wide biological profiles only differ from each other in a small subset of regions/genes. However, in Fig 2, it seems the performance of mAP is usually lower than mp-value, for small percentages of differential features. Does that mean that mAP can not efficiently dissect phenotypic activities when the number of differential features (e.g. differential expressed genes) are relatively small?

We have now updated both the process of calculating mAP p-values (calculating directly rather than aggregating AP p-values) and the protocol for running simulations used to make Figure 2 (see opening section of this letter). Specifically, we now re-use the same random seed to generate each dataset, leading to more consistent performance along the X axis as more features differ from controls. Our overall conclusion from the results presented in the updated Figure 2 and new Supplementary Figures S2-S4 is that no method performs well when only a few features differ from controls. When many features do not differ across perturbations, dimensionality reduction or feature selection methods may aid in removing these uninformative features and increasing the signal-to-noise ratio. We added this suggestion explicitly to the paragraph discussing Figure 2 (lines 297-300).

R2.4. Related to the comment above, beyond the basic evaluation of global similarities/differences across profiles, it is usually more desired to pinpoint the specific differential features in biological studies. Could mAP also prioritize such features?

This raises a very useful point. While we have an example of evaluating a feature group importance for phenotypic activity assessment on Cell Health data (Supplementary Figure S2C), extending this approach to the extreme of evaluating each feature individually would be possible to do in theory, but does not appear as a practical choice, especially in high-dimensional feature spaces. One of the core advantages of mAP is simplicity in its reliance on a distance metric specifically for multivariate similarity assessment by retrieval, whereas other approaches would

be more suitable for the important and difficult problem of identifying individual differential features (e.g., feature selection and feature importance evaluation methods).

R2.5. The permutation test for p-value calculation should be based on re-shuffling of profile labels, instead of just shuffling the ranked lists.

Because AP calculation relies solely on ranks (relative positions in the list) and not distance metric values, both these approaches are equivalent. Our null hypothesis is that M profiles in one group come from the same distribution as N profiles in the reference group (i.e. produced by the same data-generating process). During the test, it does not matter which point is chosen as a query because we always sort the rest of profiles by decreasing similarity and convert the result into the binary rank list of size $N+(M-1)$. Since the number of points is fixed and the rank list is binary, the null distribution covering all possible ranking outcomes only depends on two parameters: $M-1$ and $N+(M-1)$ and has the exact size equal to the binomial coefficient $\binom{N+(M-1)}{M-1}$. If we keep re-shuffling labels, choosing one profile as a query, sorting the rest and converting the result into a binary rank list, we will recover the same null distribution.

To check whether the mAP score is statistically significant, we employ permutation testing, asking:

what is the probability that the ranking we got is by chance?

We made sure to describe this more explicitly in the Methods section, *Assigning significance to mAP scores* (lines 673-680).

R2.6. The simulated datasets are based on normal distributions, which is very rare in real biological profiles. The authors need to benchmark the performance using additional distributions.

Also, the authors also need to test the performance based on different variances. Currently, the variances are fixed to be the same.

We thank the reviewer for the suggestion. We ran additional simulations to address this request and added their results as two new Supplementary Figures. Supplementary Figure S3

shows retrieval results using simulated data with perturbed features sampled from shifted normal distribution with variance 2. Supplementary Figure S4 shows retrieval results using simulated data with perturbed features sampled from shifted Cauchy distribution to demonstrate a case with heavy tails. We also added references to these results in the main text (lines 282-297).

R2.7. In Fig 3A, "Results are shown for a subset of an arbitrary pair of plates as a proof-of-concept". The authors need to show the complete set of results (maybe as supplementary figures if needed).

We thank the reviewer for this suggestion to improve Figure 3A. We calculated mAP on all three cell lines, in all three scenarios and for all plates. We updated the figure and also the description of these results in the main text (lines 326 through 379).

R2.8. The authors may compare mAP versus a basic approach of directly clustering the profiles based on the multi-variable features and check how separable the profiles are clustered.

As suggested, we compared mAP performance with directly clustering profiles using k-means clustering on simulated data. We showed these comparisons in Figures 2 and S2-4, and discussed them in the main text (lines 253 through 311).

R2.9. As a suggestion, I wonder whether the authors could demonstrate the advantage of mAP in the context of data sparsity, which is a common problem in single-cell profiles. If mAP can efficiently delineate the profiles even if the profiles contain highly sparse features, that would strengthen the significance.

We thank the reviewer for bringing up this challenging issue. We agree that missing values and sparsity pose an important challenge for evaluating high-dimensional profiling data and we think this issue is mainly related to data preprocessing. Different approaches to this problem can be recommended depending on levels and patterns of data sparsity. For example, one strategy is to use dimensionality reduction to identify a lower-dimensional space where features are shared across vectors. Alternatively, weighted similarity or adaptive imputation can fill in values in a way that minimizes the negative impact of sparsity, especially when its patterns are unaligned across samples. In our practice working with image-based morphological profiling¹⁶, the proportion of missing values is typically not very high overall and they are usually related to either "problematic" samples or features, which can be excluded from analyses without losing too much information, especially when dealing with single-cell data. However, there is no single rule that applies in all cases, and the best practice is to collect convincing evidence supporting these decisions. We discussed this important issue in the main text (lines 200-204).

R2.10. In lines 439-443: the 4% retrieval for nELISA seems very low. The authors may further clarify whether it is due to the limitation of mAP.

The lower retrieval percentage for nELISA is likely due to A549 cells' limited secretory capacities, the absence of immune stimulation, and a mismatch between pathways targeted by small molecules and nELISA readouts. This result can be viewed as another example of how mAP evaluation can be used to characterize experimental design (along with other examples of plate and well position effect assessment, Figure 3A). We made sure to emphasize this in the main text (lines 486-494).

R2.11. In lines 389-393: the authors briefly mentioned the issues of incomplete and imperfect profile annotations. Since mAP requires the pre-existing profile annotations/labels, it seems to be an important problem. The authors may further discuss this issue and how to overcome in practical implementations.

The issue of incomplete and imperfect profile annotations is important and pervasive in profiling. The reliance on annotations depends on the task at hand. For example, phenotypic activity assessment (whether a perturbation leads to a phenotype distinguishable from controls, Figure 1E) only depends on perturbation identity annotation, which is typically present in every experiment's metadata (i.e. which samples are replicates of the others). Similarly, metadata can be used to evaluate profile inter- vs intra-group similarities based on various block designs (see **R2.2**). The issue of incomplete and imperfect profile annotations mainly affects the task of what we call phenotypic consistency assessment (whether perturbations are grouped by a common label distinguishable from other perturbations). Here, there is no workaround: if we want to assess whether a given experimental setup yields profiles that are phenotypically consistent—showing the “correct” similarities between samples—we need some source of information about what those correct answers are. It should be noted that this issue is method-independent, as it relates to formulation of a question, rather than to means of trying to answer it—in other words, all metrics for this task face this same limitation. More broadly, recent works highlight how leveraging multiple sources of annotation, for example, combining pathway databases, can strengthen the interpretability of phenotypic profiling, helping to recapitulate known relationships and improving benchmarking outcomes. For mAP applications, incorporating diverse annotations could similarly enhance profile retrieval by allowing cross-validation of biological relationships under different contexts, such as across cell types or experimental conditions. We now explicitly note this in the main text (lines 438-443).

Remarks on code availability:

Overall the GitHub repository is well organized with clear structure and readme for installation and usage. The authors may need to provide additional notations for the scripts in "tests".

We added the README file for unit tests located in the “tests” subfolder with descriptions of tests and instructions on how to run them.

Reviewer #3 (Remarks to the Author):

R3.0 The manuscript proposes a framework based on the mean average precision (mAP) to analyze and compare high-dimensional profiles. As the authors acknowledge, mAP is a routinely used metric in machine learning, and the proposed permutation test is also standard in statistics. The proposed framework appears to be a direct application of mAP with limited novelty. Another major concern is that it's difficult to justify the use of this framework in real practice given the existence of many easier alternatives.

We thank the reviewer for constructive suggestions to improve the manuscript. This general comment is related to our response to **R2.0**, which included manuscript edits to better demonstrate the advantages of the proposed approach over established techniques. In short, we emphasized how the mAP framework is a better fit for profiling data evaluation tasks; we also extended benchmarks of mAP against existing alternatives; we highlighted how a single unified framework can be used for various tasks, profiling data and perturbation types. Please see our response to **R2.0** for specific pointers to related edits.

We specifically hope to make the mAP framework as easy to use as alternatives through the release of our open-source library that implements: (i) a flexible framework for profile grouping using metadata-based block design, and (ii) mAP calculation optimized processing large datasets (see **R3.3**). We added demo examples showing applications to real data (see **R1.5**), and a separate repository with full source code for reproducing all results in this manuscript (see **R3.7**).

R3.1. Consider a typical analysis involving two high-dimensional profiles. It is difficult to envision a scenario where the focus would be on testing between multivariate data. In most biomedical and biological applications, including those discussed in the manuscript, marginal testing of individual features is more useful. For example, in the perturbation and single-cell data analysis, why would one choose mAP over more straightforward and informative test of individual genes?

This comment is related to our response to **R2.4** above on usage of mAP for differential feature analysis:

“While we have an example of evaluating a feature group importance for phenotypic activity assessment on Cell Health data (Supplementary Figure S2C), extending this approach to the extreme of evaluating each feature individually would be possible to do in theory, but does not appear as a practical choice, especially in high-dimensional feature spaces. One of the core advantages of mAP is simplicity in its reliance on a distance metric specifically for multivariate similarity assessment by retrieval, whereas other approaches would be more suitable for the important and difficult problem of identifying individual differential features (e.g., feature selection and feature importance evaluation methods).”

We note that the primary questions mAP aims to help solve are usually upstream of differential feature analyses as they aim to establish a more general characterization of a dataset at hand: (a) whether perturbed profiles exhibit a phenotype distinguishable from controls (phenotypic activity, Figure 1E), and (b) whether perturbations that share annotations are distinguishable from other perturbations. These assessments may provide insights into the overall quality of the dataset/experiment and also narrow down the sample space for more detailed downstream analyses such as identification of differential features. Methods to diagnose which individual features (genes in mRNA profiles, or image features in morphology profiles) are important (for distinguishing a perturbation or group of perturbations from negative controls or from other perturbations) and are certainly needed but as the reviewer notes, this is a different problem not suited to mAP.

R3.2. In the application to the perturbation data, the manuscript demonstrates that the mAP can be used assess batch effects. However, the manuscript does not discuss why this method is better than conventional methods, for example, using a clustering analysis after dimensionality reduction or a direct differential test with respect to the batch effect. Both methods are much more straightforward and easier to interpret than the propose mAP.

Our recent publication³⁷ performed an in-depth evaluation of different batch effect assessment metrics and correction methods for morphological cell profiling. One high-level observation was that no single metric could fully evaluate batch effect correction, especially because data integration requires balancing two goals: removing batch effects and preserving biological variation. In that evaluation³⁷, we included mAP as a metric to evaluate preservation of biological signal after batch effect correction by calculating mAP retrieval of perturbation profiles against controls and against other perturbations – and we believe this metric best captures the most common biological goal: retrieving samples similar to a query sample. mAP metrics demonstrated good agreement with other metrics, but also were simpler to compute compared to alternatives that involved clustering (with the potential need for parameter tuning and/or repeat initializations) and therefore scaled better to large datasets (Supplementary Figure 7 in ³⁷). Our results presented in this manuscript (Figure 3A) demonstrate that mAP can be used as a general framework to assess various kinds of technical variation in data by taking a block design approach to grouping profiles in a way that disentangles particular variable contributions to retrieval performance (e.g. batch, plate, well position, etc.). In practice, however, multiple metrics can provide a more comprehensive assessment of the batch-integration quality. We now explicitly mention this in the main text when discussing results in Figure 3A (lines 377-379).

R3.3. The mAP framework relies on permutation to obtain statistical significance. For each set of permuted data, a pairwise distance matrix needs to be calculated. Such an approach will be computationally intensive when sample size is large. It may not be feasible in single-cell studies with tens of thousands of cells.

This is an important concern as profiling datasets grow larger with recent technological advancements. We approached it from two different perspectives. One was choosing to use a rank-based mAP statistic, which has an underlying distribution of possible permutations that only depends on the size of two groups and not on the distance values. This means that only ranking matters for calculating the associated p-value and therefore we do not compute the full distance matrix, because it's enough to permute the rank list itself to obtain the null distribution (please also see our response to **R2.5** above). Second, we invested effort in making the code efficient to run on large datasets. For example, we actively use batching and multi-threading for computing distances and caching null distributions for calculating p-values given the number of positives and negatives. This allows us to scale our mAP implementations to run on the “cpg0016[orf]” dataset that contains ~80,000 profiles (analysis shown in **Supplementary Figure S3B**). When running on single-cell data, we use additional strategies such as subsampling of negative examples (controls or other perturbations) to reduce computational demands.

R3.4. There is a rich body of literature in statistics discussing multivariate tests, such as Hotelling’s test and kernel tests. However, there is a lack of proper discussion and comparison with these methods.

We thank the reviewer for ensuring proper discussion and comparison with existing methods.

While indeed Hotelling’s T^2 test can be used for comparing multivariate means between two groups, it assumes normality and is sensitive to small sample sizes and high-dimensionality, as previously discussed in ¹² and ¹⁴. It can also be noted that our baseline method mp-value relies on Mahalanobis distance calculation that is related to Hotelling’s T^2 test, which can be viewed as the scaled version of the Mahalanobis distance between the sample mean and the population mean. While kernel methods, e.g., mean maximum discrepancy test (MMD), are non-parametric and can deal with high-dimensional data, they generally work better with larger sample sizes. I.e., MMD estimates the discrepancy between distributions, and small sample sizes (such as 2-4 replicate profiles per sample as in Figure 2) may not provide enough information to reliably estimate this discrepancy. Thus, for tasks like phenotypic activity, where the number of replicates per perturbation is often $\ll 10$, and phenotypic consistency, where each label includes anywhere between 2 and few dozen perturbations, such tests may still suffer from high variance and/or low statistical power, making it difficult to detect differences even if they exist. We added the MMD test to our benchmarking experiment using simulated data and included results in the updated Figure 2 and Supplementary Figures S2-5. We also extended discussion of existing methods (lines 70-77) and discussion of benchmarking results (lines 282-294, 307-311).

R3.5. In the comparison between CellProfiler and DeepProfiler data, the authors reported varying mAP values and retrieval rates. How to evaluate the performance of mAP from these values? What mAP value is considered large or small?

We found that the numerical range of calculated mAP scores varies across datasets, making universal interpretation of what mAP scores to consider large or small challenging. Specifically, the range depends on the size of the groups that are being compared, and thus, on the size of the null distribution (e.g., if the control group is very large even lower mAP values may already be in the 95th percentile of random mAP distribution). Therefore, we suggest performance assessment to be done using p-values as an indicator of not having enough evidence to reject the alternative hypothesis. In most datasets, we found a good correlation between mAP scores and their corresponding p-values, i.e. data points filtered out by p-value threshold have mAP values in the lower part of the range. For comparison between two alternatives, mAP scores can be used directly if group sizes match, but in general we rely on comparing percent retrieved, which is the percentage of data points with significant p-values. We made the definition of percent retrieved more explicit, please see the next response **R3.6** below.

R3.6. The definition of the retrieval rate is unclear.

We made sure to emphasize the definition of *percent retrieved* in the *mAP calculation and statistical significance* subsection of Results (lines 239-241), in the *Assigning significance to mAP scores* subsection of Methods (lines 685-686), and in figure captions.

R3.7 Remarks on code availability: The software is made available. However, the code and data for reproducing the analysis are not available.

We added an example Jupyter notebook demonstrating how to calculate mAP for the assessment of both phenotypic activity and consistency:

<https://github.com/cytomining/copairs/tree/main/examples>. We also released source code for reproducing all analyses in the manuscript in a separate repository:

https://github.com/carpenter-singh-lab/2024_Kalinin_mAP.

Dear reviewers:

We thank you all once again for your time and efforts in helping us improve the manuscript. We have reviewed the remaining comments and substantially revised the manuscript to address them. Our point-by-point responses are given below. Corresponding changes to the manuscript text are highlighted in yellow. Numerical references in this letter correspond to citation numbers in the body of the revised manuscript.

Reviewer #3 (Remarks to the Author):

The revised manuscript has addressed some of the questions I previously raised; however, two major concerns remain inadequately addressed.

We apologise for not communicating clearly enough the intended purpose and utility of the mAP framework. We are not proposing the framework as a method to identify differential features nor as a comprehensive batch effect metric, and some of our original language misled in that direction. Instead, mAP is designed for what is broadly referred to as profiling analysis, which deals with high-throughput perturbational data and evaluates signatures as holistic entities (profiles), akin to the approach taken in the Connectivity Map [4, 6]. Specifically, we focus on profile-level properties of such data: *phenotypic activity* (how distinguishable is this perturbation from negative controls?) and *phenotypic consistency* (how distinguishable is this group of perturbations from other groups?). In the detailed responses below, we hope to better articulate the utility of using mAP framework in the context of these questions.

R3.1.1 My first major concern relates to the advantage of mAP over methods designed to identify individual differential features, such as genes. Reviewer 2 raised a similar question, and while the authors' response provides some clarification, the utility of mAP in this context remains unclear. It is true that comparing high-dimensional data and identifying individual differential features are distinct computational problems. However, the manuscript fails to present a compelling argument as to why mAP analysis would become a widely adopted upstream analysis for differential feature analysis.

We appreciate the reviewer's feedback and would like to clarify our position.

The reviewer raises a concern that we were presenting the mAP framework as having advantages over differential feature analysis (DFA) methods. The reviewer also correctly notes that DFA and profiling analysis address different problems:

- DFA identifies individual features (e.g., genes) that differ between conditions, supporting downstream tasks such as pathway enrichment or network analysis [20].
- Profiling analysis (including our framework) treats high-dimensional profiles holistically to measure sample-level similarities and differences. We added a new subsection *Results: Profile evaluation as information retrieval* to state goals of profiling analysis more clearly:

“A fundamental goal of profiling analysis is to identify biologically meaningful relationships between samples by comparing their phenotypic signatures. One important application of this is the ability to annotate previously uncharacterized perturbations by comparing them to a reference dataset of annotated profiles (or “compendium”)^{1,2,4,6}. For example, a compound with an unknown mechanism of action (MoA) can be compared against a compendium of compounds with known MoAs. If the unknown compound exhibits a phenotypic signature highly similar to those of compounds targeting a specific pathway, it suggests a shared mechanism and potential therapeutic relevance.”

We confirm that the mAP framework is a method for profiling analysis and is not an alternative for or prerequisite to DFA methods. While it’s technically feasible to apply DFA methods to identify individual features for each perturbation separately and use them to infer differences between whole profiles, this approach is indirect and more cumbersome in a high-throughput setting. Directly comparing entire profiles in the common high-dimensional feature space (as mAP does) is conceptually simpler and more aligned with the profiling paradigm. We updated the Introduction to describe the motivation behind profiling analysis approaches and added a note to explicitly indicate the difference between objectives of DFA and profiling analysis methods (lines 71-73, 96-101, 341-344). We also added a new subsection *Results: Profile evaluation as information retrieval* to clarify which profiling analysis tasks we focus on and described in more detail our motivation for addressing these tasks using an information retrieval approach (lines 120-162, Supplementary Table 1).

R3.1.2 An important question to address is what additional biological insights can be gained from conducting mAP analysis, especially if it cannot replace feature-based analysis. Clarifying this point is crucial for demonstrating the broad applicability of the method.

Having established the overall goals of profiling analysis, we can now illustrate specific biological insights enabled by applications of the mAP framework to profiling data:

First, the mAP framework allows us to identify dataset-scale issues by assessing whether perturbations can reliably retrieve their own replicates and separate them from controls (phenotypic activity). For example, if most perturbations or those expected to produce strong phenotypic changes fail to be retrieved (percent retrieved is close to 0%), this may indicate widespread assay insensitivity, suboptimal feature extraction, or uncontrolled experimental variation affecting the entire dataset. Ideally, in a high-throughput experiment with diverse perturbations, we expect to observe mAP values over the whole range [0,1], which corresponds to perturbation effects varied from weak to very strong. We highlight this application in Results in more detail (lines 146-148, 280-281, 395-398).

Second, the mAP framework provides a systematic way to identify and filter out inactive or non-reproducible perturbations based on their replicate retrieval performance, thereby improving downstream analyses like clustering. By removing perturbations with poor replicability or weak activity in these retrieval tasks and by ranking the rest, we can focus subsequent

analyses on perturbations with clear, reproducible phenotypic effects. Because mAP is an aggregate of individual profile-level AP scores, we can also assess intra-group variability and identify and remove outlier profiles (or even individual cells, when done on a single-cell level). We highlight this application with edits in Results (lines 141-145, 436-448, 591-600) and in the updated Figure 3C.

Third, the framework can help identify functional similarities and differences across groups of perturbations by retrieving perturbations that share expected biological annotations (phenotypic consistency). It can confirm known relationships (if perturbations with shared biological roles cluster together, this reinforces confidence in their relatedness) and aid in discovering new connections (phenotypic signatures with high similarity may reveal previously unrecognized relationships). Perturbation-level AP scores can again be helpful in identifying specific inconsistent perturbations. We highlight this application in Results (lines 148-153, 470-474) and in the updated Figure 3F.

Finally, the framework enables direct optimization of experimental and analytical parameters specifically for phenotypic profiling applications by measuring how different choices affect retrieval performance. This provides guidance for maximizing the ability to detect and group similar biological activities in profiling datasets. We also highlight this application in Results (lines 406-418, 443-448).

To summarize, our mAP framework is a sample-level profiling analysis tool designed for efficient activity and similarity assessment of high-throughput, high-dimensional, large-scale perturbational data. Rather than replacing individual feature analysis, mAP aims to prioritize perturbations and relationships between them on the profile level, reducing the likelihood of spurious findings from noisy or weak signals, and improving efficiency of subsequent analysis.

R3.2 My second concern pertains to mAP's utility for evaluating batch effects compared to existing, more straightforward methods. The authors referred to their recent preprint on bioRxiv, arguing that "no single metric could fully evaluate batch effect correction." While this may be true, it does not establish that mAP is generally a better tool for assessing batch effects.

First, the preprint referenced primarily focuses on image-based cell profiling, as suggested by its title, "Evaluating batch correction methods for image-based cell profiling." This restricts its scope and does not address batch effect correction across diverse data types. Second, there does not appear to be any direct comparison between mAP and existing batch effect evaluation methods in the preprint's main figures. Third, the authors mentioned Supplementary Figure 7 in the preprint as containing such a comparison, but the supplementary file is unavailable on its bioRxiv page.

Rather than relying on a reference to this preprint, which has a different focus, it would be far more convincing to include a direct comparison between mAP and other

established batch effect evaluation methods within this manuscript. This would provide readers with clearer evidence of mAP's utility and advantages.

We apologize for the imprecise initial response and the confusion caused by the missing figure in the preprint.

We would like to emphasize that the mAP framework is not specifically designed for assessing batch effects and we do not claim in the paper that it is generally better than existing metrics. A direct comparison with batch effect metrics would not be appropriate here given that our framework serves a different goal; our initial response was confusing in this respect. To clarify, we use the example in Figure 3A to demonstrate that the mAP framework flexibly allows accounting for the influence of technical variation in the data *on the retrieval tasks of interest* (phenotypic activity) that it is designed to evaluate. Specifically, how sharing the same plate or well position can impact the ability to retrieve replicates of each perturbation against negative controls. That is not a comprehensive batch effect assessment, because batch effects can be manifested in ways not captured by this retrieval task, for example, a systematic shift in feature distributions across batches that affects all samples equally, making perturbations within a batch appear biologically more similar than samples across batches. If batch effects are properly corrected, the success of replicate retrieval against controls may improve, but comprehensively measuring batch effects is not limited to this task and is not the framework's purpose.

In light of this, we have revised the manuscript to make our intent clearer. Specifically, we have refined the terminology to avoid overgeneralization: in Section “Results: mAP captures diverse properties of real-world morphological profiling data...” and Figure 3A, we now describe our analysis as focused on “assessing plate and well-position effects *on phenotypic activity*” rather than broadly implying “assessing batch effects” (lines 379 through 430, 663-667). This more accurately reflects the task-specific context in which technical confounder effects are evaluated in the manuscript and avoids unintended implications about the scope of mAP for comprehensive batch effect evaluation.

We apologize for referring the missing supplementary figure about computational efficiency in our preprint (now published in Nature Communications [40], Supplement available at: https://static-content.springer.com/esm/art%3A10.1038%2Fs41467-024-50613-5/MediaObjects/41467_2024_50613_MOESM1_ESM.pdf). In this revision, we instead performed a new, detailed analysis of computational efficiency of our framework (that motivated further improvements), as proposed by the reviewer in section **R3.3**.

We hope these clarifications address the reviewer's concern and ensure the manuscript's focus remains appropriately framed.

R3.3. The authors acknowledged that computational time on large datasets could be a concern, and they have improved the efficiency of the code. It would be helpful to report the current computational time needed for each dataset in the manuscript to give readers

a rough sense. A discussion of the theoretical computational complexity will be even better.

We agree that specific runtime analysis can be helpful to potential software users, so we added a new section on Runtime analysis to the Supplemental Materials and referred to it in the manuscript (lines 623-625, 668-670). Specifically, we added a new analysis of theoretical computational complexity of mAP calculation by breaking this process down into three main stages and estimated the complexity of underlying operations for each of them. Then we also measured and reported computation time needed for mAP calculation times on each dataset in the manuscript. While testing calculations on larger data, we also noticed opportunities for additional optimizations that we implemented in the new version of our Python package: <https://github.com/cytomining/copairs/releases/tag/v0.5.1>. The new version offers substantial speed-ups at the profile grouping stage (25-300x). We also added a new notebook example, demonstrating the effect of the null size selection on the mAP p-value calculation: https://github.com/cytomining/copairs/blob/main/examples/null_size.ipynb.

We thank the reviewer for encouraging this analysis and improvements that can help easier adoption and wider impact of our work.

Dear Reviewer #3:

Thank you again for your time and efforts in helping us improve the manuscript. We have reviewed the remaining comment and revised the manuscript to address it. Numerical references in this letter correspond to citation numbers in the body of the revised manuscript.

Reviewer #3 (Remarks to the Author):

The authors' new responses have resolved my questions about the differential feature analysis and computational time. Regarding the batch effects, the authors clarified that mAP is not specifically designed for assessing batch effects. It would be helpful to clarify in the manuscript what the suggested pipeline is when batch effects are expected to be present in the data.

Should users apply another batch effect removal method first before applying mAP, or is there a way to first evaluate whether mAP can handle the batch effects in the specific case?

Given that batch effects are a common challenge in high-throughput profiling experiments, we agree that users should first apply a dedicated batch-effect correction method prior to running mAP analysis. Based on the comparative evaluations across multiple modalities and tasks [17–19], we suggest using Harmony [58] or Seurat v3 [59], as they strike a good balance in preserving meaningful biological structure while mitigating technical variation, making them reliable first choices for general-purpose batch correction. We added this recommendation explicitly to the revised manuscript (Methods, lines 638-645).